# REPRESENTATION LEARNING FROM INTERVENTIONAL DATA

## ABSTRACT

To learn data representations that are robust to distribution shifts, practitioners conduct interventions and collect interventional data in addition to passively collected observational data. However, even when the underlying causal model is known, existing approaches treat interventional data like observational data and ignore the causal model. Furthermore, these approaches assume a large number of interventional data points obtained through interventions that span the entire support of the intervened variable. This leads to representations that exhibit large discrepancies in predictive performance on observational and interventional data. In this paper, we first identify a strong correlation between interventional performance and adherence of the features to the statistical independence conditions induced by the underlying causal model. Then, we exploit this correlation and propose RepLIn to explicitly enforce the statistical independence during interventions. We demonstrate the utility of RepLIn across representative image classification tasks (attribute prediction on CelebA and image classification under corruption on CIFAR-10C and ImageNet-C) by modeling them as causal graphs and learning representations that are more robust to interventional distribution shifts.

## 1 INTRODUCTION

We consider a data-generating process that can be modeled using directed acyclic graphs (DAGs) called causal graphs. The nodes in these graphs are random variables that usually equate to semantic concepts such as the color of an object, the quantity of sugar in the blood, and the age of a person. Causal modeling allows us to intervene on one or more of these variables and observe the effects on its/their descendants. The data collected through this procedure is referred to as *interventional data*. Interventional data has traditionally been used in problems such as causal discovery and A/B testing (see **??**). Incorporating causal information into the training stage of a model finds applications such as learning disentangled representations (Locatello et al., 2019; Brehmer et al., 2022), domain generalization (Mahajan et al., 2021), and adversarial training (Zhang et al., 2021).

Several works implicitly use interventional data without considering the statistical independence relations[1] entailed during interventions. Ignoring these independence relations will result in representations that are susceptible to distribution shifts. For example, deep feature reweighting (DFR) (Kirichenko et al., 2022) proposed to retrain the classifier layer using a dataset that was balanced to break spurious correlations. To obtain this dataset, we require perfect interventions spanning the entire support of the intervened variable. However, it may not be possible to intervene with values spanning the entire support in practice. In addition, the number of interventional points available during training may be far less compared to cheaply obtained observational data.

We first consider a case study in which we observe a correlation between accuracy drop due to interventional distribution shift and dependence between features during interventions. Then we propose **rep**resentation **l**earning from **in**terventional data (RepLIn) to enforce the independence relations from the interventional causal graph during training to improve the robustness against interventional distribution shift. We demonstrate the advantage of our proposed method when interventional support is different from that during test time by comparing it against deep feature reweighting (Sec. 3). We further confirm the utility of RepLIn on face attribute classification (Sec. 4.2)

---

[1]We refer to "statistical independence" as simply "independence" for the rest of the paper

and label-dependent image corruption (Sec. 4.3). In classifying corrupted images using pretrained ImageNet models, we improve upon our baselines by $\sim \mathbf{2-4\%}$ with only 10% interventional data.

To summarize, our contributions are:

- We demonstrate a correlation between accuracy drop due to interventional distribution shift and dependence between interventional features (Sec. 2.1).

- We demonstrate that explicitly enforcing independence between interventional features minimizes the drop in accuracy under interventional distribution shifts (Sec. 2.3).

- We demonstrate the effectiveness of the proposed method over classifier fine-tuning when the interventional distribution does not match the testing distribution (Sec. 3).

## 2 THE LEARNING FROM INTERVENTIONAL DATA PROBLEM

We now formally define the learning problem of interest in this paper, namely the *learning from interventional data*, in general terms, and examine a specific case study in Sec. 2.1. The problem comprises an attribute of interest $B$ and a directed acyclic graph $\mathcal{G}$ denoting the causal relations between $B$ and its corresponding parents $\mathbf{Pa}_B = \{A_1, \dots, A_n\}$. These attributes along with other unobserved variables $U$, generate the data $X$, i.e., $X = g_X(B, A_1, \dots, A_n, U)$. Intervention on $B$ breaks

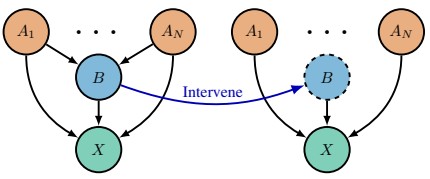

Figure 1: Causal graphs during observation (left) and intervention on $B$ (right)

the statistical dependence on its parents, i.e., now $B^{\text{int}} \perp\!\!\!\perp \mathbf{Pa}_B$, as shown in Fig. 1. By intervention, we refer to *hard intervention* defined in Peters et al. (2017), where the variable $B$ is set to a specific value, drawn from a known distribution. Note that we do not require any knowledge of the other unobserved nodes in this causal graph. For training, data samples from both the observational distribution and the interventional distribution are available, i.e., $\mathcal{D}^{\text{obs}} \sim P(X^{\text{obs}}, B^{\text{obs}}, A_1^{\text{obs}}, \dots, A_n^{\text{obs}})$ and $\mathcal{D}^{\text{int}} \sim P(X^{\text{int}}, B^{\text{int}}, A_1^{\text{int}}, \dots, A_n^{\text{int}})$. Given $(\mathcal{D}^{\text{obs}}, \mathcal{D}^{\text{int}})$ and $\mathcal{G}$, the goal is to predict $B$ and $A_i$ from attribute-specific representations $F_B = f_B(X)$ and $F_{A_i} = f_{A_i}(X)$ respectively.

### 2.1 DOES INTERVENTIONAL ACCURACY CORRELATE WITH STATISTICAL INDEPENDENCE?

First, we consider a motivating case study on a synthetic dataset and establish a relation between predictive performance on interventional data and statistical independence between the corresponding attribute features under intervention. Then, building upon this observation, we propose RepLIn, a simple yet effective solution to learn representations that are robust to *intervention-induced distribution shifts* by exploiting interventional data.

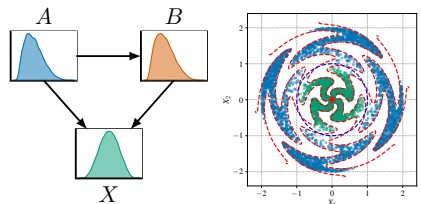

(a) Observational graph and data

**Case Study:** Consider the causal graph shown in Fig. 2(a). Here, $A$ and $B$ are binary random variables that generate the observed real-valued data $X$. $X$ is also affected by unobserved noise variables[2]. $A$ itself could be a function of external random factors which are unobserved and of no interest to us. However, the distribution of $B$ is only affected by $A$, as denoted by the arrow between them. In Fig. 2(b), we intervene on $B$ and thus induce a change in its distribution, i.e., an intervention-induced distribution shift. Since the intervention is independent of $A$, intervened $B$ is also independent of $A$, denoted by removing the arrow between $A$ and $B$. The analytical

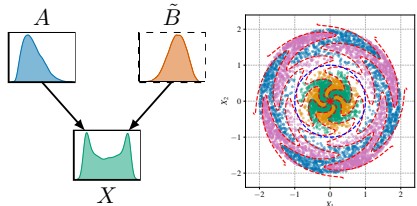

(b) Interventional graph and data

Figure 2: **WINDMILL Dataset:** $A$ and $B$ are binary random variables that are causally linked to each other and to $X$ as shown in (a). By intervening on $B$ as shown in (b), we make $A \perp\!\!\!\perp B$.

---

[2]We skip the noise variables in our illustrations for simplicity.

relations between $A$, $B$ and $X$ during observation and intervention are:

| Observation | Intervention | |
|---|---|---|
| $A \sim \text{Bernoulli}(0.6)$ | $A \sim \text{Bernoulli}(0.6)$ | $X := g_X(A, B)$ |
| $B := A$ | $B \sim \text{Bernoulli}(0.5)$ | |

The equations in blue govern the observational distribution and those in red govern the interventional distribution. The function that generates $X$ from $A$ and $B$ is unaffected by interventions. Following (Peters et al., 2017), $:=$ indicates the causal assignment operator. Visually, the samples look like a windmill. The value of $A$ determines the blade of the windmill, and $B$ determines the radial distance. In order to make the data more stochastic, the precise angle and radial distance of the points are sampled from an unobserved distribution independent of $A$ and $B$. To make the data more challenging, we shear each blade according to a sinusoidal function of the radial distance. The task here is to accurately predict $A$ and $B$ from $X$ at test time. We construct $g_X$ such that $A$ and $B$ are fully recoverable from $X$. The exact mathematical formulation is provided in App. H.

**Training:** We have $N$ samples for training in total where $\beta N$ are interventional and $(1 - \beta)N$ are observational with $0 < \beta < 1$ typically being a small value. For this demonstration, we set $N = 40000, \beta = 0.1$. Therefore, we have 36000 observational and 4000 interventional samples. We train a feed-forward network with three hidden layers to extract features $F_A$ and $F_B$ corresponding to $A$ and $B$, respectively. Following the standard ERM framework, the cross entropy error in predicting $A$ and $B$ from $F_A$ and $F_B$ provides the training signal. Fig. 3(a) and Fig. 3(b) show the accuracy of ERM in predicting $A$ and $B$ on observational and interventional data during validation.

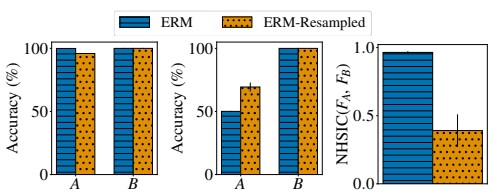

(a) Observation (b) Intervention (c) Dependence

Figure 3: The gap in performance correlates well with a gap in the measure of dependence of the learned features on interventional data.

*Ideally, we expect no drop in accuracy of $A$ from observation to intervention if the model does not learn the spurious correlation between $A$ and $B$.* However, we observe that ERM barely performs better than random chance in predicting $A$ on interventional data. As a remedy, we consider a stronger version of ERM by reweighing the interventional data by resampling it as often as the observational data. We refer to this version as "ERM-Resampled". Now the model sees interventional batches $\left(\frac{1-\beta}{\beta}\right)$-times as many observational batches. The equivalent loss for a learning function $f$ now transforms to $\mathcal{L}_{\text{total}}(f, X) = \sum_{i=1}^{N_{\text{obs}}} \mathcal{L}_{\text{pred}}(f, X_i^{\text{obs}}) + \left(\frac{1-\beta}{\beta}\right) \sum_{i=1}^{N_{\text{int}}} \mathcal{L}_{\text{pred}}(f, X_i^{\text{int}})$. Although ERM-Resampled performs better than vanilla ERM, there is still a large gap between accuracy in predicting $A$ on observational and interventional data.

## 2.2 MEASURING STATISTICAL DEPENDENCY BETWEEN INTERVENTIONAL FEATURES

A key characteristic of perfect interventions on causal graphs is that the variable being intervened upon becomes independent of all its nondescendants. As such, we hypothesize that *if the features corresponding to the intervened variable are more statistically independent of the features corresponding to its nondescendants, then the predictive accuracy of the nondescendants of the intervened variables will be less affected by interventions.*

**Dependence Measure:** To measure dependence between a pair of high-dimensional continuous random variables $P$ and $Q$, we use HSIC (Gretton et al., 2005), a kernel-based measure of dependency. Given $N$ i.i.d. samples $\left\{P^{(i)}\right\}_{i=1}^{N}$ and $\left\{Q^{(i)}\right\}_{i=1}^{N}$ from $P$ and $Q$, HSIC between $P$ and $Q$ can be computed as $\text{HSIC}(P, Q) = \frac{1}{(N-1)^2} \text{Trace}\left[\boldsymbol{K}_P \boldsymbol{H} \boldsymbol{K}_Q \boldsymbol{H}\right]$, where $\boldsymbol{H}$ is the $N \times N$ centering matrix, $\boldsymbol{K}_P \in \mathbb{R}^{N \times N}$ is a Gram matrix whose entry at the $i$-th row and $j$-th column is $k_P\left(P^{(i)}, P^{(j)}\right)$, where $k_P(\cdot, \cdot)$ is the kernel function associated to a given universal kernel (e.g., RBF kernel). $\boldsymbol{K}_Q$ is defined similarly. Since HSIC is unbounded, following (Li et al., 2021), we consider a normalized HSIC score (NHSIC) defined as $\text{NHSIC}(P, Q) = \frac{\text{HSIC}(P,Q)}{\sqrt{\text{HSIC}(P,P) \, \text{HSIC}(Q,Q)}}$.

We use the NHSIC metric to compare the statistical dependence between the features in the WIND-MILL problem. Fig. 3(c) shows the difference in NHSIC values between the features $F_A$ and $F_B$ from interventional data. We observe that features learned with ERM-Resampled are more independent than those learned by vanilla ERM. Dependence between features from interventional data indicates that they share information even though the random variables they are associated with are independent. We conjecture that it might be due to $F_A$ learning from $B$, since $B$ is a spurious feature for $A$ during observations.

## 2.3 REPLIN: ENFORCING STATISTICAL DEPENDENCY ON INTERVENTIONAL FEATURES

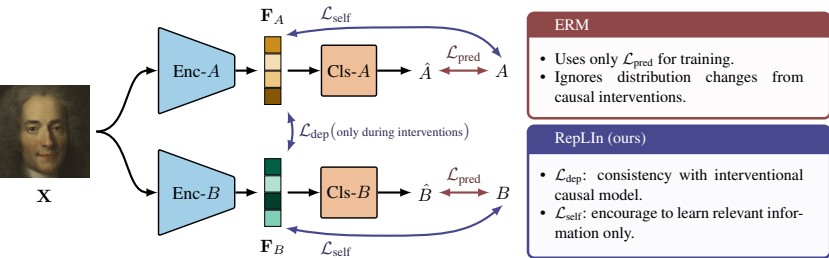

Figure 4: Schematic illustration of **RepLIn** for a causal graph with two attributes ($A \rightarrow B$) and $X = f(A, B, U_X)$. Encoders (Enc-A, Enc-B) learn representations ($F_A, F_B$) corresponding to each label, which is then used by their corresponding classifiers (Cls-A, Cls-B) for prediction. On interventional samples, we minimize $\mathcal{L}_{\text{dep}}$ between the features to ensure their independence. On all samples, we minimize $\mathcal{L}_{\text{self}}$ to encourage the representations to only learn relevant information.

As noted in the previous subsection, neither ERM nor ERM-Resampled explicitly ensures that the features adhere to the same relations as their latent variable counterparts during interventions. As a result, we also observed that there is a correlation between interventional accuracy and interventional feature dependence. Based on this observation, we propose RepLIn to explicitly enforce the same causal relations between the features during interventions as the latent variables. We hypothesize that enforcing this independence will force the model to learn features that are robust to interventional distribution shifts.

To enforce independence between interventional features, we propose to use dependence-guided regularization denoted as $\mathcal{L}_{\text{dep}}$ over the prediction loss function (cross-entropy for classification tasks) used in ERM. We refer to this regularization as "dependence loss" and is defined for the general case in Sec. 2 as $\mathcal{L}_{\text{dep}} = \frac{1}{n} \sum_{i=1}^{n} \text{NHSIC}(F_{A_i}^{\text{int}}, F_B^{\text{int}})$ , where the superscript "int" denotes features extracted from interventional samples, i.e., we seek to minimize the dependence loss *only* for the interventional samples in our training set.

However, $\mathcal{L}_{\text{dep}}$ alone is insufficient since the features can take a shortcut and simply learn irrelevant features and minimize $\mathcal{L}_{\text{dep}}$. To avoid such pathological scenarios and encourage the model to only learn relevant information, we introduce another loss that maximizes the dependency between a feature and its corresponding label. We employ this "self-dependence loss" on both observational and interventional data and define it as $\mathcal{L}_{\text{self}} = 1 - \frac{\text{NHSIC}(F_B, B) + \sum_{i=1}^{n} \text{NHSIC}(F_{A_i}, A_i)}{2(n+1)}$ .

In summary, RepLIn optimizes the following total loss: $\mathcal{L} = \mathcal{L}_{\text{pred}} + \lambda_{\text{dep}} \mathcal{L}_{\text{dep}} + \lambda_{\text{self}} \mathcal{L}_{\text{self}}$ , where $\lambda_{\text{dep}}$ and $\lambda_{\text{self}}$ are weights that control the contribution of the respective losses. A pictorial overview of RepLIn is shown in Fig. 4.

## 3 CLASSIFIER FINETUNING MAY NOT BE ENOUGH

Classifier finetuning emerged recently as a potential solution to spurious correlations (Menon et al., 2020; Kirichenko et al., 2022; Rosenfeld et al., 2022; Qiu et al., 2023). The foundation of such approaches is that learned representations contain both invariant and spurious features, and with the help of a fine-tuning dataset, the classifier can be retrained to rely on only the invariant features. However, practitioners may be limited in providing a fine-tuning dataset that spans the entire support

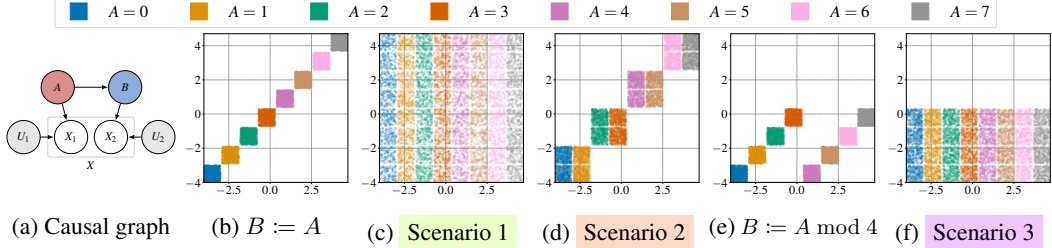

| (a) Causal graph | (b) $B := A$ | (c) Scenario 1 | (d) Scenario 2 | (e) $B := A \bmod 4$ | (f) Scenario 3 |

Figure 5: **When interventional support does not match test support:** The latent covariates $A$ and $B$ affect each other and generate data $X$ according to the causal graph in (a). During observation ((b) & (e)), $A$ and $B$ are correlated, making it difficult for the model to learn the true decision boundaries. Interventional data that matches the test distribution (c) can help. However, the interventional support may not always match that of the test distribution ((d)& (f)). Removing spurious information entirely is desirable in these settings.

of the intervened variable. For example, we cannot change the medicinal dose for critically ill patients to study the effect of the said medicine on vitals. Also, the quantity of interventional data available during training may not be sufficient to build the fine-tuning dataset. We argue that, under such circumstances, it is advisable to remove spurious information from the representations entirely.

To support our argument, we generate a synthetic dataset consisting of two categorical random variables $A$ and $B$ with eight classes each. During observation, $A$ and $B$ are causally linked as $A \rightarrow B$. Their relationship during observation can be mathematically written as $A \sim P_A; B := A$, where $P_A$ is the uniform categorical distribution over eight classes. By intervening on $B$, it takes value from an interventional distribution $P_B^I$, where $I$ denotes an intervention from a class of interventions $\mathcal{I}$. The input data $X = \begin{bmatrix} X_1 \\ X_2 \end{bmatrix}$ are generated from $A$, $B$, and noise variables $U_1, U_2$ according to the causal graph shown in Fig. 5(a) as $X_1 = g_{X_1}(A, U_1)$ and $X_2 = g_{X_2}(B, U_2)$.

Consider the observational data shown in Fig. 5(b). Clearly, there are infinitely many classifiers that have zero risk on the observational data but non-zero risk on the test distribution shown in Fig. 5(c). To learn the true classifier, half of the training dataset is obtained through interventions. The class of interventions $\mathcal{I}$ comprises of the following: **Scenario 1:** (Fig. 5(c)) interventional support is same as the domain of $B$ (full support), **Scenario 2:** (Fig. 5(d)) interventional support correlates with $A$ (partial support), **Scenario 3:** (Fig. 5(f)) interventional support changes between training and testing (different support). Observational data corresponding to scenarios 1 and 2 are shown in Fig. 5(b) and that corresponding to scenario 3 is shown in Fig. 5(e).

We use a linear layer with ReLU on top to extract features and train a linear classifier with these features. In each scenario, we train a model using ERM, classifier finetuning (ClsFT), and RepLIn. Both ERM and ClsFT train their models by minimizing classification error (e.g., cross-entropy) on the training data. Once the training is complete, ClsFT fine-tunes the classifier layer using a fine-tuning dataset made from the interventional data. Every experiment is repeated ten times.

| Scenario | ERM | ClsFT | RepLIn | Scenario | ERM | ClsFT | RepLIn |
|----------|-----|-------|--------|----------|-----|-------|--------|
| Fig. 5(c) | $100.00 \pm 0.00$ | $100.00 \pm 0.00$ | $100.00 \pm 0.00$ | Fig. 5(c) | - | - | - |
| Fig. 5(d) | $99.92 \pm 0.08$ | $99.96 \pm 0.06$ | $98.37 \pm 1.49$ | Fig. 5(d) | $48.36 \pm 2.75$ | $48.39 \pm 2.67$ | $66.91 \pm 10.33$ |
| Fig. 5(f) | $99.52 \pm 0.24$ | $99.74 \pm 0.20$ | $99.58 \pm 0.31$ | Fig. 5(f) | $81.57 \pm 10.58$ | $81.45 \pm 12.16$ | $94.00 \pm 1.71$ |

| (a) Accuracy on seen support | (b) Accuracy on unseen support |

Table 1: Although ERM and ClsFT perform well on seen support, their accuracy diminishes on unseen support. However, RepLIn suffers a smaller accuracy drop on unseen support. Refer to App. C for more observations.

**Observations:** Tab. 1a and Tab. 1b compare the results of ERM, ClsFT and RepLIn on seen and unseen supports respectively. When an interventional dataset with the same support and distribution as during test time is available (Fig. 5(c)), all methods achieve zero error on the entire support. In this scenario, there is no unseen support. When the support during intervention correlates with $A$ during training (Fig. 5(d)), both ERM and ClsFT show a significant drop ($\sim 52\%$) in their performance on unseen region compared to seen regions. However, RepLIn suffers a smaller drop in

performance ($\sim 33\%$). Accuracy drop can be observed in scenario 3 (Fig. 5(f)) as well, where the support during training and testing are completely different. Surprisingly, all methods suffer smaller drops in accuracy compared to the former scenario. ERM and ClsFT have $\sim 19\%$ higher misclassification rate on unseen support compared to seen, while RepLIn shows only $\sim 6\%$ drop in accuracy. App. C analyzes RepLIn further by comparing its decision boundaries with those of the baselines.

## 4 EXPERIMENTAL EVALUATION

In this section, we evaluate the performance and generality of RepLIn in comparison to the ERM baselines across three scenarios corresponding to different causal data-generating mechanisms and associated interventions. These include the WINDMILL dataset introduced in Sec. 2.1, facial attribute prediction on CelebA, and robustness to image corruptions on CIFAR-10C and ImageNet-C. Our experiments are designed to validate the following hypothesis: **Q1)** *Is there a strong correlation between accuracy on interventional data and statistical independence of the features corresponding to the intervened variable.*, and **Q2)** *Does explicitly minimizing the dependence between features on interventional data improve interventional accuracy.*

**Training Hyperparameters and Baselines:** A detailed description of the training settings for each experiment, along with the corresponding hyperparameters, can be found in App. D. We note that the value of $\lambda_{\text{dep}}$ and $\lambda_{\text{self}}$ is kept fixed across all proportions of interventional data $\beta$. For all experiments, we consider standard ERM and ERM-Resampled (Chawla et al., 2002; Cateni et al., 2014; Idrissi et al., 2022) as our baselines.

**Evaluation Criterion:** Our primary interest is in investigating the prediction accuracy of variables that are unaffected during interventions. Ideally, if the learned features respect causal relations, we expect to see no change in the prediction accuracy of variables corresponding to the parents of the intervened variable in the causal graph. Since we optimize NHSIC during training, we rely on another measure of independence, namely kernel canonical correlation (KCC) (Bach & Jordan, 2002) to evaluate the dependence between the features on interventional data during testing. We repeat each experiment five times with different random seeds and report the mean and standard deviation as a shaded region in plots.

### 4.1 WINDMILL DATASET

We first verify our method on the synthetic dataset that helped us identify the relation between the performance gap in predicting $A$ on observational and interventional data in Sec. 2.1. As a reminder, the causal graph consists of two binary random variables $A$ and $B$, where $A \rightarrow B$. During interventions, we manually set $B$ to randomly chosen values, breaking the dependence between $A$ and $B$. Earlier, we showed that ERM and ERM-Resampled fail when $\beta$ takes very small values. We vary $\beta$ from 0.5% to 50% and compare RepLIn against ERM and ERM-Resampled. We consider an additional baseline "Dep-on-all", where we naively minimize $\mathcal{L}_{\text{dep}}$ and $\mathcal{L}_{\text{self}}$ on *all* samples. All methods share the same architecture. We observed that adding an extra dimension and normalizing the features to a unit sphere improved performance (App. E).

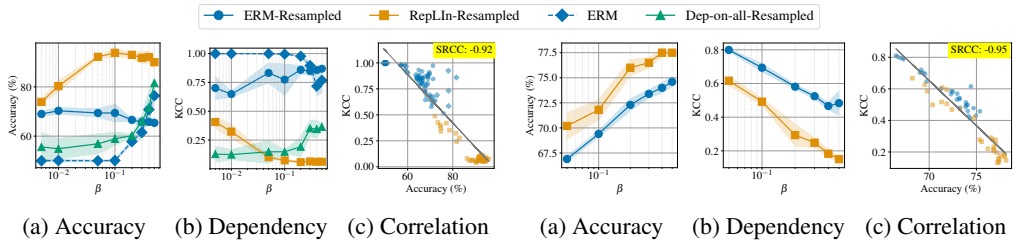

| (a) Accuracy | (b) Dependency | (c) Correlation | (a) Accuracy | (b) Dependency | (c) Correlation |

Figure 7: Results on WINDMILL dataset.  Figure 8: Results on Facial Attribute Prediction

Fig. 6(a) compares the interventional accuracy of $A$ as a function of the amount of interventional data ($\beta$). We observe that our model outperforms both ERM and ERM-Resampled on all values of $\beta$. RepLIn outperforms Dep-on-all, indicating that *naively enforcing independence on all samples*

*is suboptimal*. Furthermore, when 50% of the total data is interventional, ERM-Resampled still outperforms vanilla ERM, suggesting that the improvement could be due to treating the data as separate batches of observational and interventional samples only, in addition to resampling. We also compare the dependence between the features on interventional data in Fig. 6(b). Again, observe that explicitly enforcing independence on interventional features during training indeed minimizes dependence on unseen interventional data during testing. Fig. 6(c) plots the interventional accuracy and KCC between the features of each run of each method. To confirm our hypothesis in Sec. 2.1, we should obtain a Spearman rank correlation coefficient (SRCC) (Spearman, 1904) of -1. We estimate SRCC from the data to be -0.92, which strongly supports our hypothesis. We demonstrate visually in App. A that the representations learned by RepLIn are less affected by interventional shifts.

## 4.2 FACIAL ATTRIBUTE PREDICTION

We verify the utility of RepLIn for predicting facial attributes on the CelebA dataset (Liu et al., 2015). CelebA dataset is provided with 40 labeled attributes. We consider two of these attributes – smiling and gender – as random variables affecting each other causally.

Although the true underlying relation between smile and gender is unknown, we adopt the resampling procedure by Wang & Boddeti (2022) to induce a desired causal relation between the attributes (smile → gender) and obtain samples. Consequently, in this scenario, the causal relationship between the attribute labels is known. Specifically, to simulate this causal relation, we sample smile first and then sample gender according to a conditional probability distribution over smile. We then sample an image whose attribute labels match the sampled values. We treat the diversity in the images as a result of unobserved latent variables.

Given the face images, we first extract features from ResNet18 (He et al., 2016) pre-trained on ImageNet (Deng et al., 2009). Then, similar to the architecture for WINDMILL experiments, we employ a shallow MLP to act on the features, followed by a linear classifier to predict the attributes. Our loss functions act upon the features of the MLP. We use 30,000 samples for training and 15,000 for testing. The causal model for this experiment and some sample images are shown in Fig. 9.

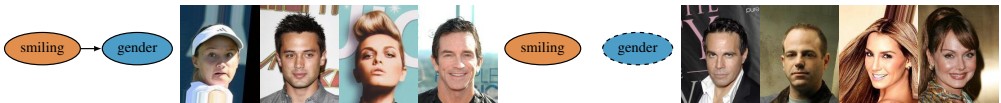

(a) Observational causal graph and samples generated from it

(b) Interventional causal graph and samples generated from it

Figure 9: Causal model for CelebA before and after intervention along with sample images from these models

Fig. 8 reports the experimental results of ERM-Resampled and RepLIn as a function of the amount of interventional data. We make the following **observations**: 1) as the amount of interventional data increases, the interventional prediction accuracy of both methods improve, 2) across all proportions of interventional data, RepLIn consistently outperforms the baseline by about 2%-4%, and 3) interventional accuracy and KCC show strong negative correlation (SRCC=-0.95). At the same time, the dependency between $F_A$ and $F_B$ is significantly lower than the baselines as $\beta$ increases. Attention maps corresponding to these predictions are shown in App. B.

## 4.3 ROBUSTNESS TO IMAGE CORRUPTION

Here we consider a scenario with a three-variable causal graph. We construct a causal model for label-dependent corruption as shown in Fig. 10(a). We consider ten possible corruption types from (Hendrycks & Dietterich, 2019) (e.g., Gaussian noise, frost), which are chosen based on the label. The chosen corruption is applied to a clean image to obtain our input corrupted image. Our goal is to predict the class label on interventional data. As part of RepLIn, we also predict the noise type but do not evaluate its accuracy since it is not a variable of interest.

In this case, spurious correlations would correspond to relying on the type of noise as a proxy for predicting the image label. We obtain the interventional images by intervening in the type of corruption, making the choice of corruption independent of the label. This setup bears similarity

to the one considered in (Zhang et al., 2020). However, unlike our task, where the noise is label-dependent in observational data, they only consider label-independent image augmentation since their goal is to learn models that are invariant to augmentation changes at test time.

**Learning from Scratch:** We consider CIFAR-10C (Hendrycks & Dietterich, 2019) with five choices of image corruption and learn RepLIn model end-to-end from raw images. The network includes a CNN to extract features and MLPs on top of these features to extract attribute-specific features. Our dependency and self-dependency loss functions act on these attribute-specific features.

We present the results in Fig. 8. We make the following **observations**: 1) as expected, interventional accuracy of all methods improves with $\beta$, i.e., access to more interventional data at training; 2) explicitly enforcing independence on features for interventional data leads to consistent accuracy gains over ERM-Resampled, and 3) features from unseen interventional data are more statistically independent for RepLIn, especially as $\beta$ increases. In summary, our results indicate that 1) modeling label-dependent corruptions as causal models can overcome spurious correlations in data, and 2) explicitly enforcing independence constraints on the learned features leads to appreciable performance gains over ERM-Resampled.

**Transfer Learning from Pre-Trained Features:** Next, we evaluate the pre-trained feature extractors that cover a wide range of architectures, datasets, and training schemes. We use open-sourced pre-trained models from (Wightman, 2022) and (Meta, 2022). Specifically, we consider (1) **ResNet50** trained using standard supervised learning (He et al., 2016), (2) ResNet50 trained using **MoCoV2** algorithm (Chen et al., 2020), (3) **VIT-B/32** trained in a supervised fashion on ImageNet-21K (Dosovitskiy et al., 2020), (4) VIT-B/32 used as backbone in **CLIP** (Radford et al., 2021) trained on a 2-billion image subset of LAION-5B (Schuhmann et al., 2022) and then fine-tuned on ImageNet-21K.

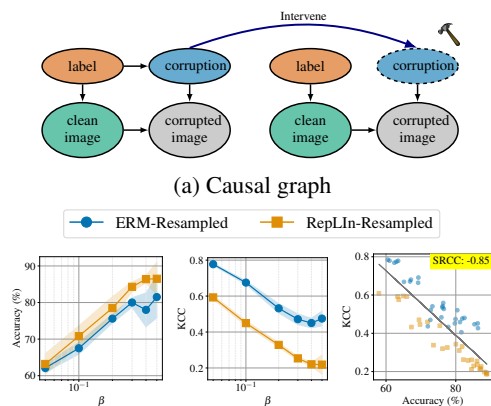

(a) Causal graph

(b) Accuracy  (c) Dependence  (d) Correlation

Figure 10: **Robustness against label-dependent corruption:** (a) shows the underlying causal model and (b) and (c) compare ERM-Resampled and RepLIn. (d) shows the correlation between feature dependence and accuracy during interventions.

By being pre-trained on larger datasets or with different loss functions, these models may inherently exhibit robustness to the noise corruption model considered in Fig. 10(a). For RepLIn and the baselines, we introduce a shallow MLP over the backbone feature extractor and predict the class label. For RepLIn, we apply our loss functions to the MLP's features. In this experiment, we evaluate on ImageNet validation set with randomly applied corruptions. We assign each of the 1000 classes a corruption through the causal graph. All the images from the class will have the assigned corruption applied to them. The support of the interventional data varies similar to the scenarios in Sec. 3.

| Method | $\beta$ | Full support | | | |
|---|---|---|---|---|---|
| | | R50 | MoCoV2 | ViT | CLIP |
| ERM | 0.5 | $57.17 \pm 0.12$ **(-0.84)** | $35.49 \pm 0.04$ **(-2.66)** | $51.26 \pm 0.08$ **(-0.98)** | $48.26 \pm 0.03$ **(-0.64)** |
| ClsFT | | $57.16 \pm 0.04$ **(-0.85)** | $36.82 \pm 0.08$ **(-1.33)** | $51.10 \pm 0.11$ **(-1.14)** | $48.36 \pm 0.08$ **(-0.54)** |
| RepLIn | | $\mathbf{58.02 \pm 0.07}$ | $\mathbf{38.15 \pm 0.04}$ | $\mathbf{52.24 \pm 0.07}$ | $\mathbf{48.90 \pm 0.08}$ |
| ERM | 0.9 | $51.18 \pm 0.14$ **(-2.80)** | $28.11 \pm 0.09$ **(-4.16)** | $37.35 \pm 0.19$ **(-2.51)** | $36.65 \pm 0.18$ **(-1.81)** |
| ClsFT | | $45.74 \pm 0.16$ **(-8.24)** | $19.99 \pm 0.15$ **(-12.29)** | $16.56 \pm 0.32$ **(-23.31)** | $18.23 \pm 0.19$ **(-20.23)** |
| RepLIn | | $\mathbf{53.98 \pm 0.13}$ | $\mathbf{32.27 \pm 0.13}$ | $\mathbf{39.86 \pm 0.18}$ | $\mathbf{38.46 \pm 0.10}$ |
| | | Partial support | | | |
| | | R50 | MoCoV2 | ViT | CLIP |
| ERM | 0.5 | $54.70 \pm 0.04$ **(-1.19)** | $33.72 \pm 0.05$ **(-1.65)** | $47.22 \pm 0.19$ **(-1.70)** | $45.50 \pm 0.11$ **(-1.35)** |
| ClsFT | | $54.28 \pm 0.07$ **(-1.62)** | $33.12 \pm 0.08$ **(-2.25)** | $46.83 \pm 0.13$ **(-2.10)** | $45.58 \pm 0.12$ **(-1.27)** |
| RepLIn | | $\mathbf{55.90 \pm 0.06}$ | $\mathbf{35.37 \pm 0.08}$ | $\mathbf{48.93 \pm 0.08}$ | $\mathbf{46.85 \pm 0.04}$ |
| ERM | 0.9 | $50.82 \pm 0.08$ **(-2.16)** | $28.41 \pm 0.08$ **(-1.86)** | $37.16 \pm 0.19$ **(-2.31)** | $36.61 \pm 0.15$ **(-1.81)** |
| ClsFT | | $44.67 \pm 0.12$ **(-8.31)** | $19.99 \pm 0.11$ **(-10.29)** | $17.27 \pm 0.27$ **(-22.20)** | $19.76 \pm 0.31$ **(-18.65)** |
| RepLIn | | $\mathbf{52.98 \pm 0.11}$ | $\mathbf{30.28 \pm 0.02}$ | $\mathbf{39.47 \pm 0.17}$ | $\mathbf{38.41 \pm 0.20}$ |
| | | Different support | | | |
| | | R50 | MoCoV2 | ViT | CLIP |
| ERM | 0.5 | $53.18 \pm 0.05$ **(-0.54)** | $32.05 \pm 0.04$ **(-0.95)** | $45.32 \pm 0.05$ **(-0.39)** | $39.79 \pm 0.10$ **(-0.47)** |
| ClsFT | | $52.64 \pm 0.08$ **(-1.08)** | $31.81 \pm 0.07$ **(-1.20)** | $44.67 \pm 0.17$ **(-1.04)** | $39.43 \pm 0.11$ **(-0.82)** |
| RepLIn | | $\mathbf{53.72 \pm 0.09}$ | $\mathbf{33.00 \pm 0.03}$ | $\mathbf{45.71 \pm 0.14}$ | $\mathbf{40.26 \pm 0.08}$ |
| ERM | 0.9 | $50.12 \pm 0.14$ **(-1.37)** | $28.09 \pm 0.03$ **(-1.84)** | $36.09 \pm 0.12$ **(-1.51)** | $32.26 \pm 0.08$ **(-0.83)** |
| ClsFT | | $43.81 \pm 0.25$ **(-7.68)** | $20.45 \pm 0.31$ **(-9.48)** | $16.02 \pm 0.46$ **(-21.57)** | $17.45 \pm 0.09$ **(-15.64)** |
| RepLIn | | $\mathbf{51.49 \pm 0.06}$ | $\mathbf{29.93 \pm 0.11}$ | $\mathbf{37.60 \pm 0.17}$ | $\mathbf{33.09 \pm 0.11}$ |

Table 2: **Results on ImageNet-C:** RepLIn outperforms ERM-Resampled and ClsFT by a significant margin, especially when the proportion of interventional data available is very little.

We use a 100,000 subset of ImageNet (Deng et al., 2009) as our training set and consider two settings - one with 10% interventional data and another with 50% interventional data. Tab. 2 shows the image

classification results on interventional data. We observe that RepLIn outperforms the baselines for all considered backbones, proportions of interventional data, and intervention support types.

We also make the following **observations**: 1) Each method performs its best when the interventional support matches that of the test distribution, 2) ClsFT performs significantly worse than ERM-Resampled when the amount of interventional data is limited, and 3) Comparing the methods during full support intervention and $\beta = 0.5$, RepLIn shows most improvement on MoCoV2 and least improvement on CLIP – both backbones trained using contrastive loss while the former was trained solely on images while the latter was trained on image-text pairs.

## 5 RELATED WORK

**Learning using Interventional Data:** Interventional data is key in causal discovery (Lippe et al., 2021; Yu et al., 2019; Ke et al., 2019; Wang et al., 2022; He & Geng, 2008) as one can only retrieve causal relations up to Markov equivalent graph without interventions or assumptions on the causal model. For example, known interventional targets have been used for unsupervised causal discovery of linear causal models (Subramanian et al., 2022), interventional and observational data have been leveraged for training a supervised model for causal discovery (Ke et al., 2022), and interventions with unknown targets were used for differentiable causal discovery (Brouillard et al., 2020). Unlike this paper, these approaches are neither concerned with representation learning, and since the causal graph is unknown, the interventional and observational data are treated equally. Interventional data also find applications in reinforcement learning (Gasse et al., 2021; Ding et al., 2022) and recommendation systems (Krauth et al., 2022). Interventional data has also been leveraged for identifiable causal representation learning. Refer to Appendix G for a detailed review.

**Training with Data Imbalance:** In many practical scenarios, there is a heavy imbalance between the amount of observational and interventional samples at hand for learning. In such cases, resampling the data according to the inverse sample frequency is effective in improving generalization to the minority class. Recent approaches such as MAPLE (Zhou et al., 2022), dynamic importance reweighting (Fang et al., 2020) and SRDO (Shen et al., 2020) also *learn to resample* using a separate validation set that acts as a proxy for the test set. However, such learned resamplers require access to a large validation dataset that reflects the interventional distribution, which is not always practically feasible. Recent studies (Idrissi et al., 2022; Gulrajani & Lopez-Paz, 2020) have shown that ERM with simple resampling is a strong baseline for spurious correlations and domain generalization. Therefore, we propose an approach that is agnostic to data imbalance while still leveraging the underlying statistical property that distinguishes interventional from observational data.

## 6 CONCLUSION

This paper considered the problem of learning from observational and interventional data by leveraging the knowledge of the statistical properties induced by interventions in the underlying data-generating process. First, we established a strong correlation between interventional accuracy and statistical dependence between features on interventional data. Building on this observation, we proposed RepLIn to mimic the true underlying causal relations by explicitly enforcing statistical independence between features on interventional data. We showed that explicitly enforcing statistical independence between features during intervention is preferable to merely fine-tuning the classifier on the interventional data. Experimental evaluation of RepLIn across different scenarios corresponding to different causal graphs has shown that RepLIn is able to improve predictive accuracy across differing proportions of interventional data consistently. Finally, we modeled corrupted image classification as a causal graph and leveraged RepLIn to learn image features that are more robust under interventions to image corruption.

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

In our main paper, we identified a correlation between interventional accuracy and dependence between interventional features and developed RepLIn that exploited this correlation for robust predictions during interventions. Here, we provide some additional analysis to support our main results. The appendix is structured as follows:

## A   DISTRIBUTION OF THE LEARNED REPRESENTATIONS

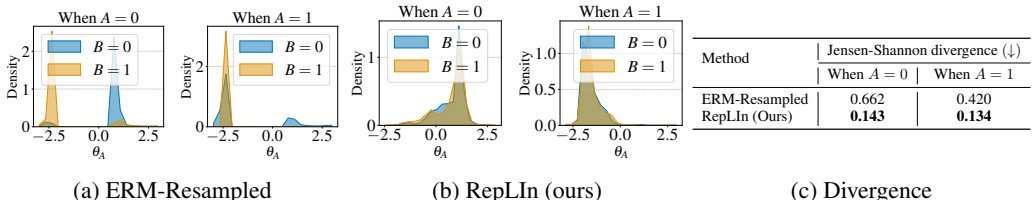

| Method | Jensen-Shannon divergence ($\downarrow$) | |
|---|---|---|
| | When $A = 0$ | When $A = 1$ |
| ERM-Resampled | 0.662 | 0.420 |
| RepLIn (Ours) | **0.143** | **0.134** |

(a) ERM-Resampled      (b) RepLIn (ours)      (c) Divergence

Figure 11: Feature visualization for ERM-Resampled (left) and RepLIn (center) on the WINDMILL dataset. (right) Jensen-Shannon divergence between $P(F_A^{\text{int}}|B = 0, A = a)$ and $P(F_A^{\text{int}}|B = 1, A = a)$, which ideally should be zero when intervening on B.

We compare the features learned by ERM-Resampled and RepLIn on WINDMILL dataset to gain a better understanding of what they actually learn. Since the features are normalized, we visualize the polar angle as histograms. Specifically, we are interested in the histogram of $F_A$ for a fixed value of $A$ and changing values of $B$. If the features are robust, they should not change with $B$. From the visualization in Fig. 11, we note that features from RepLIn are more robust to interventional distribution shifts than those from ERM.

## B   ATTENTION MAPS FOR FACIAL ATTRIBUTE PREDICTION

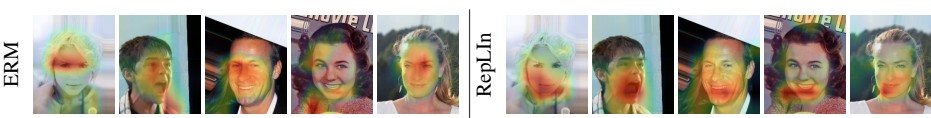

Figure 12: Image regions that contribute to predicting `smile`.

Since our features on CelebA are high-dimensional, we employ Grad-CAM (Selvaraju et al., 2017) to analyze the features and compare them against those learned by resampled-ERM. Since our primary metric is accuracy in predicting `smile` during interventions, we visualize the parts of the input image that the models attend to for predicting a smile. Fig. 12 shows the attention maps when trained with 10% interventional data. Observe that RepLIn tends to focus more on the region around the lips while resampled-ERM attends to other regions of the face too.

# C  COMPARING LEARNED MODELS FROM ERM, CLSFT, AND REPLIN

In this section, we compare the models learned using ERM, ClsFT, and RepLIn to gain an insight into why their performances differ. To that end, we compare the decision boundaries of these models, particularly noting the misclassified regions.

**Setup:** In the setup that we introduced in Sec. 3, we considered two variables of interests $A$ and $B$. They are categorical random variables that can take eight classes. These variables, along with the unobserved noise variables $U_1, U_2$, generate the input signal $X$. During observation, these variables are causally linked. By intervening on $B$, we break their causal relation. Since there are several classifiers that can achieve zero-error on the observational data alone, we use interventional data for training. Precisely, 50% of the training data comes from interventions. Refer to Fig. 5 for visualization of the causal relations and the data points.

**Decision boundaries:** As mentioned earlier, we look closely at the decision boundaries to gain understanding about what each method learns. In every case, the true decision boundaries are formed by parallel vertical lines. In each decision boundary, the misclassified points are shown using black markers. Samples from the training dataset – observational and interventional points – are shown in color denoting their classes.

We considered three scenarios for intervention. We describe them below along with the discussion on the learned decision boundaries in that scenario.

**Full support:** In this scenario, the interventional support matches that of the test distribution, i.e. full support. This is the most ideal scenario since the model sees samples from all possible combinations of $A$ and $B$. Fig. 13 compares the decision boundaries of ERM, ClsFT and RepLIn. Since the interventional data seen during training are uniformly sampled from the entire support, a model with sufficient capacity can learn the true decision boundary. We observe that all methods are able to achieve zero-error classification.

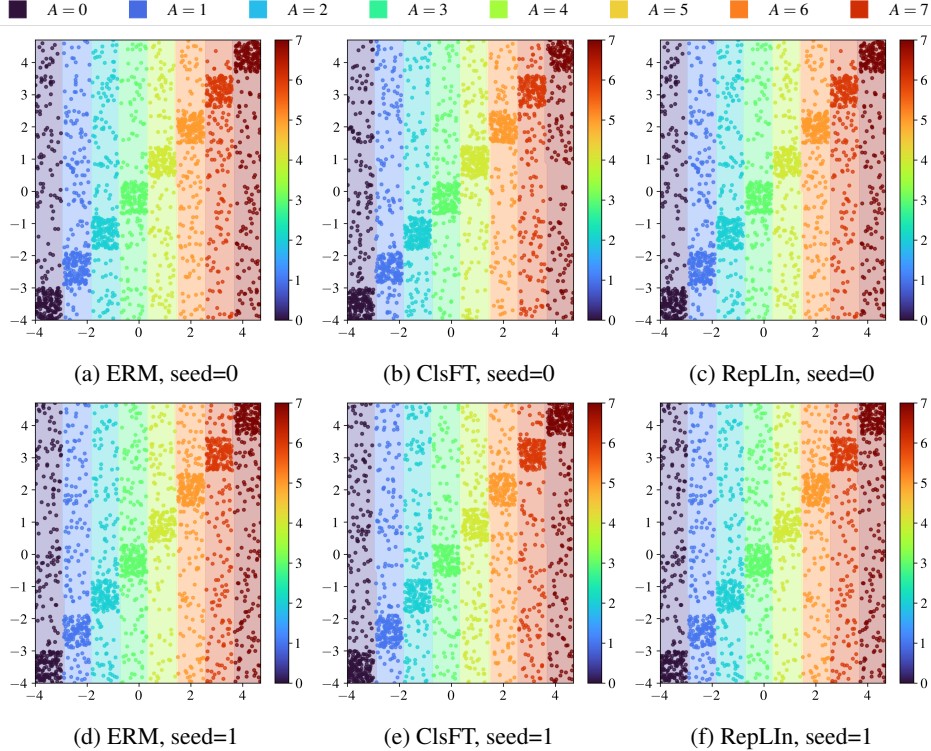

Figure 13: Comparing decision boundaries of ERM, ClsFT and RepLIn for two seeds (each row) when the interventional support matches that of test distribution ( Scenario 1 )

**Partial support:** In this scenario, the interventional support depends on the value of $A$, i.e. partial support. Fig. 14 compares the decision boundaries of ERM, ClsFT and RepLIn. Even with the interventional data, there are clearly infinite zero-error classifiers for the training data. Since ERM and ClsFT optimize to minimize error only on the seen points, their models can converge to one of these classifiers. However, RepLIn enforces a stronger statistical independence regularizer on the model. Therefore, our models learn decision boundaries which are *closer* to the optimal decision boundaries. The result of this difference in approach can be seen in the decision boundary between $A = 3$ and $A = 4$ in Fig. 14(c). RepLIn learns a more vertical (hence, closer to the true) decision boundary at the expense of a few misclassified points in the training set.

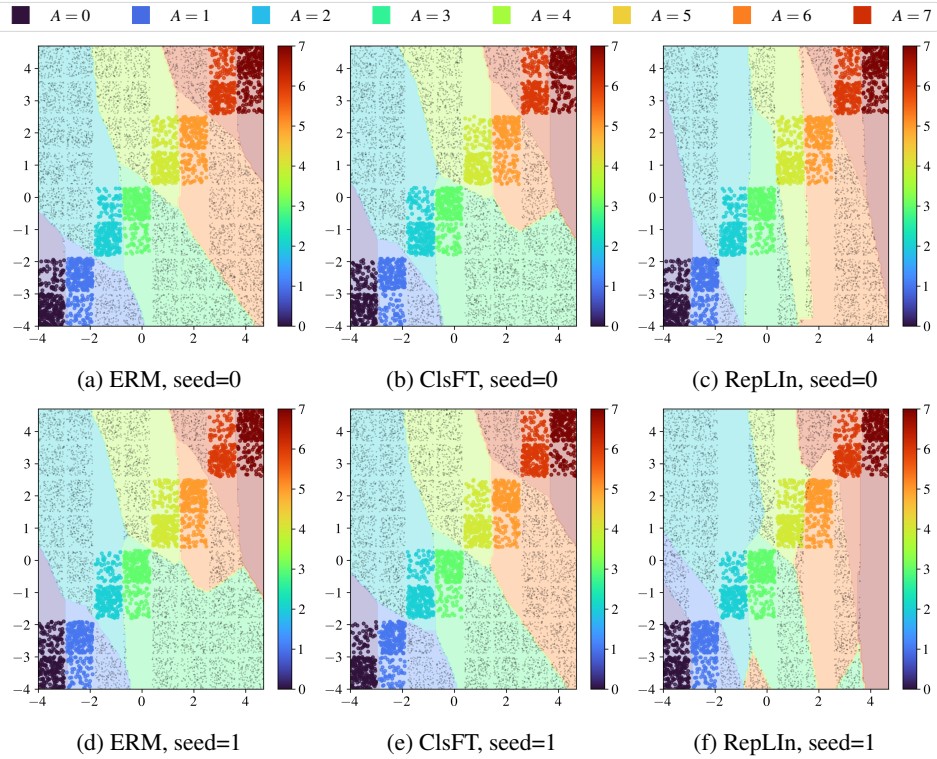

Figure 14: Comparing decision boundaries of ERM, ClsFT and RepLIn for two seeds (each row) when the interventional support depends on the value of $A$ ( Scenario 2 )

**Different support:** In this scenario, the interventional support is completely different during training and testing, i.e. different support. Fig. 15 compares the decision boundaries of ERM, ClsFT and RepLIn. As mentioned before, both ERM and ClsFT minimize error on seen data, while RepLIn minimizes statistical dependence for stronger regularization. As a result, ERM and ClsFT achieve zero error on the training set but exert little control over the decision boundary in regions of unseen support. On the other hand, RepLIn exploits the training data better to learn the true decision boundary.

# D IMPLEMENTATION DETAILS

We implement our models using PyTorch (Paszke et al., 2019) and use Adam (Kingma & Ba, 2014) as our optimizer with its default settings. Common hyperparameters shared ERM baselines and RepLIn (such as number of data points, number of epochs, etc.) are shown in Tab. 4. Other hyperparameters specific to RepLIn are shown in Tab. 3. For training stability, we warm up $\lambda_{\text{dep}}$ from 0 to its set value between $\texttt{start}N$ and $\texttt{end}N$ epochs where $N$ is the total number of epochs, and $\texttt{start}$ and $\texttt{end}$ are fractions shown in Tab. 3.

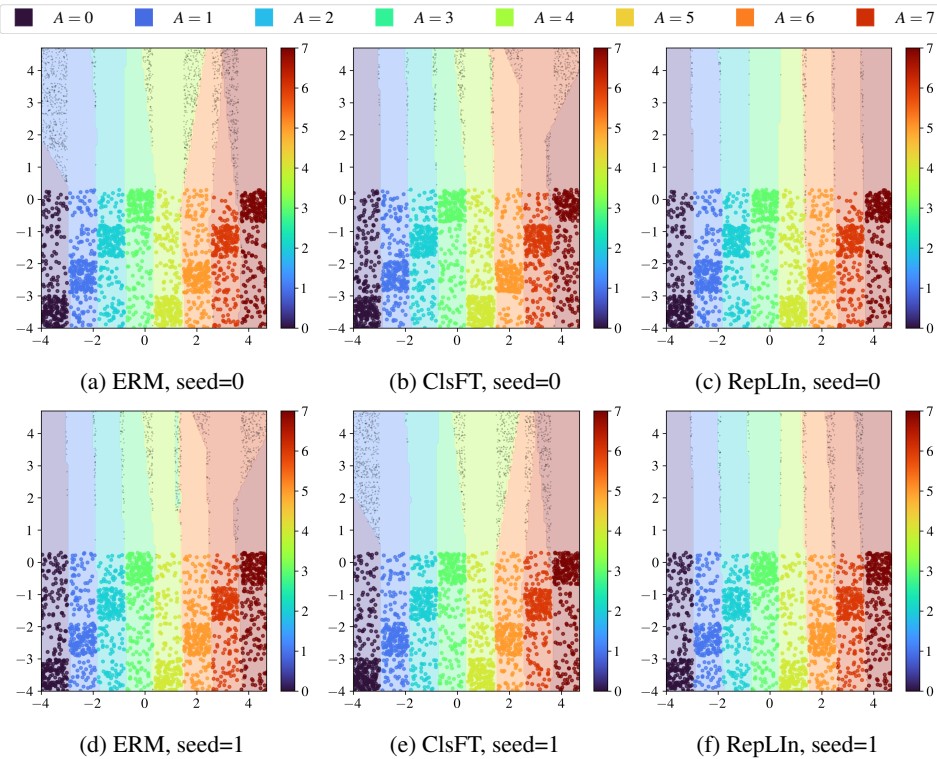

Figure 15: Comparing decision boundaries of ERM, ClsFT and RepLIn for two seeds (each row) when the interventional support is different during training and testing ( Scenario 3 )

Table 3: Hyperparameters for RepLIn

| Dataset | $\lambda_{\text{dep}}$ | $\lambda_{\text{self}}$ | start | end |
|---|---|---|---|---|
| WINDMILL | 10 | 1 | 0.66 | 0.99 |
| CelebA | 10 | 1 | 0.2 | 0.99 |
| CIFAR10-C | 1 | 1 | 0.4 | 0.9 |
| ImageNet-C | 1 | 1 | 0.2 | 0.99 |

Table 4: **Common hyperparameters.** For WINDMILL, we used a MultiStep(milestones=[1000]) with gamma=0.5 for ERM baselines and gamma=0.1 for RepLIn.

| Dataset | #Training samples | Epochs | Batchsize | Learning rate | Scheduler |
|---|---|---|---|---|---|
| WINDMILL | 40,000 | 3000 | 4000 | 2e-3 | See caption |
| CelebA | 30,000 | 100 | 1000 | 1e-3 | No scheduler |
| CIFAR10-C | 40,000 | 1000 | 2000 | 1e-3 | MultiStep(milestones=[50], gamma=0.5) |
| ImageNet-C | 80,000 | 300 | 2000 | 2e-3 | StepLR(step_size=100, gamma=0.5) |

For all methods, we first extract label-specific features from the inputs and pass them through a corresponding classifier to predict the label. The architecture of the feature extractor is the same for all methods on a given dataset, except on the WINDMILL dataset. The classification layer is a linear layer mapping from feature dimensions to the number of classes. The specific details for each dataset are provided below.

**WINDMILL dataset:** For ERM baselines, we use an MLP with two layers of size 40 and 1, with a ReLU activation after each layer (except the last) to extract the features. However, we observed that it was difficult to enforce independence using 1-dimensional features. Therefore, we used 2-dimensional features for RepLIn which were then normalized to lie on a circle. Essentially, the features from the baselines and RepLIn have the same intrinsic dimensionality of 1.

**CelebA dataset:** We first extract features from the raw image using a ResNet-18 (He et al., 2016) pre-trained on ImageNet (Deng et al., 2009). Although these features are not optimal for face attribute prediction, they have been shown to be useful for face verification (Sharif Razavian et al., 2014). Additionally, it makes the binary attribute prediction task more challenging. We extract attribute-specific features from this input using a linear layer that maps it to a 500-dimensional space.

**CIFAR-10-C dataset:** We train a CNN from scratch to extract features from the corrupted image. Fig. 16 shows the architecture of this CNN. An MLP with two hidden layers of dimensions 100 and 10 extracts features corresponding to the label and the corruption type from these CNN features.

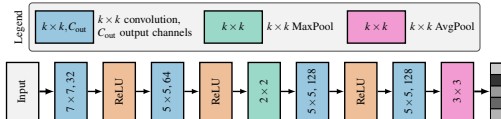

Figure 16: Architecture of the CNN used in CIFAR-10-C experiment

**ImageNet-C dataset:** We analyze the robustness of some of the commonly used image classification models pre-trained on ImageNet (Deng et al., 2009) against label-dependent corruption. Using the features extracted by these classification models as input, we extract label-specific and corruption-specific features using a linear layer with 500-dimensional output.

# E    USING UNNORMALIZED FEATURES ON WINDMILL DATASET

In our experiments on WINDMILL dataset, we observed that normalizing features helped in enforcing independence better. Fig. 17 compares the interventional accuracy and KCC between interventional features of ERM-Resampled and RepLIn - each with raw features and normalized features. For a fair comparison, they have the same architecture – the final feature dimension is 2. Without normalization, the model learned to minimize statistical dependence between interventional features at the expense of observational performance.

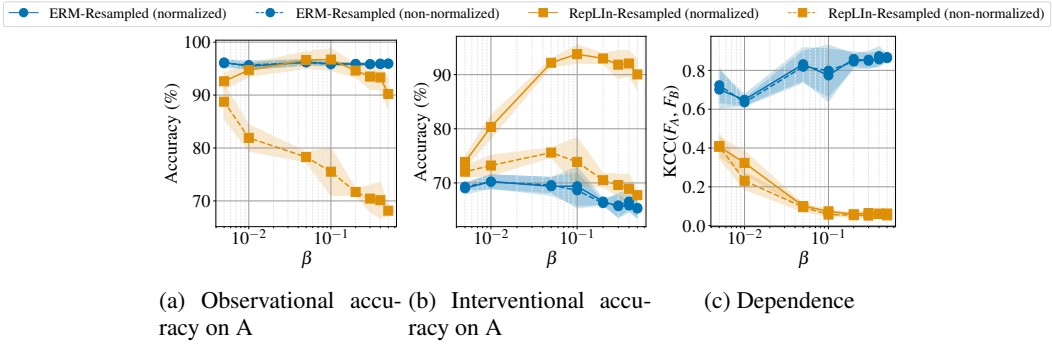

Figure 17: Advantage of normalizing the features for enforcing the independence better

# F    SIMILARITIES AND DIFFERENCES TO INVARIANT RISK MINIMIZATION SETTING

**Our setting:** Given observed data $X$, the task is to predict the labels $Y$ that generated $X$. We know that there exist causal relations between the labels that cannot be modified without intervening on one or more labels. The models are trained on a combination of observational and interventional data, where the latter is sampled from a known interventional causal graph.

**Invariant Risk Minimization (IRM) setting:** The goal of IRM (Arjovsky et al., 2019; Liu et al., 2021; Lu et al., 2021; Chevalley et al., 2022; Magliacane et al., 2018) is to predict labels $Y$ from observed data $X$, which is a function of the labels and an *environment* variable $E$ such that $E \perp\!\!\!\perp Y$.

Here, the objective is to learn a predictor that is invariant across the environments. IRM models are trained on data collected from different environments. An example of this setting is domain generalization where domains act as environments.

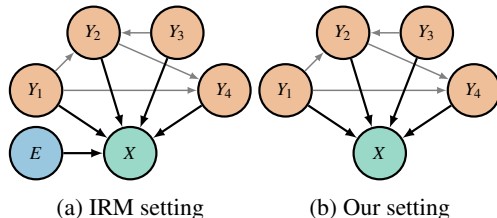

(a) IRM setting    (b) Our setting

Figure 18: Difference between the causal model in IRM setting and our setting.

**Similarities and differences:** The larger goal of both IRM and our method is to learn features that are robust to a distributional shift. However, they differ in the source of this distributional shift. Fig. 18 shows the causal graphs considered under the IRM setting and our problem setting. The distributional shift in IRM stems from the change in environment. Their training data consists of sub-datasets corresponding to different environments such as $\mathcal{D}_1 \sim P(X, Y | E = e_1)$, $\mathcal{D}_2 \sim P(X, Y | E = e_2)$, etc. The distributional shift in our setting originates from interventions. Interventional datasets can be written as $\mathcal{D}_1^{\text{int}} \sim P(X, Y | do(Y_1 = y_1))$, $\mathcal{D}_2^{\text{int}} \sim P(X, Y | do(Y_2 = y_2))$, etc. As a result, IRM is not concerned with the causal relations between labels, while we are primarily concerned with causal relations between the labels.

## G  REVIEW OF IDENTIFIABLE CAUSAL REPRESENTATION LEARNING

The primary objective of identifiable causal representation learning is to learn a representation such that it is possible to identify the latent factors (up to scale and permutation) from the representation. These methods are commonly built upon autoencoder-based approaches. The advantage of learning a causal representation is that the decoder then implicitly acts as the true underlying causal model, facilitating counterfactual evaluation and, sometimes, disentangled factors of variation.

Locatello et al. (2019); Khemakhem et al. (2020) showed that disentangled representation learning was impossible without additional assumptions on both the model and the data. Some of the inductive biases that have been proposed since to learn disentangled representations include auxiliary labels (Hyvarinen & Morioka, 2016; Hyvarinen et al., 2019; Sorrenson et al., 2020; Khemakhem et al., 2020; Ahuja et al., 2022b; Kong et al., 2022), temporal data (Klindt et al., 2021; Yao et al., 2022; Song et al., 2023), and assumptions on the mixing function (Sorrenson et al., 2020; Yang et al., 2021; Lachapelle et al., 2022; Zheng et al., 2022; Moran et al., 2022).

**Use of interventional data:** Some works also use interventional data as weak supervision for identifiable representation learning (Lippe et al., 2022b; Brehmer et al., 2022; Ahuja et al., 2022a; 2023; Varıcı et al., 2023; Varici et al., 2023; von Kügelgen et al., 2023). Lippe et al. (2022b) learns identifiable representations from temporal sequences with possible interventions at any time step. Similar to our setting, they assume the knowledge of the intervention target. They also assume that the intervention on a latent variable at a time step does not affect other latent variables in the same time step. Lippe et al. (2022a) relaxes the latter assumption as long as perfect interventions with known targets are available. Von Kügelgen et al. (2021); Zimmermann et al. (2021) showed that self-supervised learning with data augmentations allowed for identifiable representation learning. Brehmer et al. (2022) use pairs of data samples before and after some unknown intervention to learn latent causal models (LCMs). Ahuja et al. (2022a) learns identifiable representations from sparse perturbations, with identifiability guarantees depending on the sparsity of these perturbations. Sparse perturbations can be treated as a parent class of interventions where the latent is intervened through an external action such as in reinforcement learning. Ahuja et al. (2022b) use interventional data for causal learning under some assumptions on the nature of support for non-intervened variables. Varıcı et al. (2023) relax the polynomial assumption on the mixing function and proves identifiability when two uncoupled hard interventions per node are available along with observational data. Varici et al. (2023) learn identifiable representations from data observed under different interventional distri-

butions with the help of the score function during interventions. von Kügelgen et al. (2023) uses interventional data to learn identifiable representations up to nonlinear scaling. In addition to the above uses of interventional data, a few works (Saengkyongam & Silva, 2020; Saengkyongam et al., 2023; Zhang et al., 2023) have also attempted to predict the effect of unseen joint interventions with the help of observational and atomic interventions under various assumptions on the underlying causal model.

**Difference from our setting:** The general objective in identifiable causal representation learning is to "learn both the true joint distribution over both observed and latent variables" (Khemakhem et al., 2020). The objective of this work is to provide a method to learn representations that are robust to interventional distribution shifts under the assumption of known interventional targets and their parents. In other words, we are not interested in learning the joint distribution of the observed and the latent variables, but rather in developing a method to exploit data samples with known interventional targets. For example, as large models such as (Radford et al., 2021), (Brown et al., 2020), (Touvron et al., 2023) and (Dehghani et al., 2023) become more ubiquitous, efficient methods to improve these models with minimal amounts of experimentally collected data will be of interest.

## H  GENERATING WINDMILL DATASET

We provide the exact mathematical formulation of WINDMILL dataset described in Sec. 2.1. We define the following constants:

| Constants | Description | Default value |
|---|---|---|
| $n_{\text{arms}}$ | Number of "arms" in WINDMILL dataset | 4 |
| $r_{\text{max}}$ | Radius of the circular region spanned by the observed data | 2 |
| $\theta_{\text{wid}}$ | Angular width of each arm | $\frac{0.9\pi}{n_{\text{arms}}} = 0.7068$ |
| $\lambda_{\text{off}}$ | Offset wavelength. Determines the complexity of the dataset | 6 |
| $\theta_{\text{max-off}}$ | Maximum offset for the angle | $\pi/6$ |

Table 5: Constants used for generating WINDMILL dataset, their meaning, and their values.

$$R_B \sim \mathcal{B}(1, 2.5) \qquad \text{(Sample radius)}$$

$$R = \frac{r_{\text{max}}}{2}\left(BR_B + (1-B)(2-R_B)\right) \qquad \text{(Modify sampled radius based on } B)$$

$$\Theta_A \sim \mathcal{C}\left(\left\{2\pi\frac{i}{n_{\text{arms}}+1} : i = 0, \ldots, n_{\text{arms}} - 1\right\}\right) \qquad \text{(Choose an arm)}$$

$$U \sim \mathcal{U}(0, 1) \qquad \text{(To choose a random angle)}$$

$$\Theta_{\text{off}} = \theta_{\text{max-off}}\sin\left(\pi\lambda_{\text{off}}\frac{R}{r_{\text{max}}}\right) \qquad \text{(Calculate radial offset for the angle)}$$

$$\Theta = \theta_{\text{wid}}(U - 0.5) + A\left(\Theta_A + \frac{\pi}{n_{\text{arms}}}\right) + (1-A)\Theta_A + \Theta_{\text{off}}$$
$$\text{(Angle is decided by } A \text{ and the radial offset)}$$

$$X_1 = R\cos\Theta \qquad \text{(Convert to Cartesian coordinates)}$$
$$X_2 = R\sin\Theta$$

$$X = \begin{bmatrix} X_1 \\ X_2 \end{bmatrix}$$

PyTorch code to generate WINDMILL dataset is provided in Listing 1.

## I  PYTORCH CODE TO GENERATE TOY DATA WITH CHANGING INTERVENTIONAL SUPPORT

PyTorch code to generate the dataset used in Sec. 3 is shown in Listing 2.

Listing 1: Code for WINDMILL dataset

```python
import math
import torch

# Constants
num_arms = 4 # number of blades in the windmill
max_th_offset = 0.5236 # max offset that can be added to the angle for shearing (= pi/6)
r_max = 2 # length of the blade
num_p = 20000 # number of points to be generated
offset_wavelength = 6 # adjusts the complexity of the blade

# Sample latent variables according to the causal graph.
A = torch.bernoulli(torch.ones(num_points) * 0.6)
if observational_data:
    B = A
else:
    B = torch.bernoulli(torch.ones(num_points) * 0.5)

# Convert A, B to X.
th_A0 = torch.linspace(0, 2*math.pi, num_arms+1)[:-1]
th_A1 = torch.linspace(0, 2*math.pi, num_arms+1)[:-1] + math.pi/num_arms
# Choose a random arm for A=0 from possible arms. Likewise for A=1.
th_A0 = th_A0[torch.randint(num_arms, (num_p,))]
th_A1 = th_A1[torch.randint(num_arms, (num_p,))]

# beta distribution with alpha=1, beta=3
beta_dist = torch.distributions.beta.Beta(1, 2.5)

# Sample r according to B. If B=0, sample a small r, else sample a large r.
# r ranges from 0 to r_max
B0_r = beta_dist.sample(torch.Size([num_p])) * r_max/2.
B1_r = r_max - beta_dist.sample(torch.Size([num_p])) * r_max/2.
r = B * B0_r + (1-B) * B1_r

# Sample theta according to A.
# Choose the theta arm according to A and then sample from this using a uniform
    distribution.

# First we will have a cartwheel style.
theta = torch.rand(num_p)*th_wid + th_A0*(1-A) + th_A1*A - th_wid/2.

# Add an offset to theta according to r.
th_offset_mod = torch.sin((r/r_max)*offset_wavelength*math.pi)
th_offset = max_th_offset*th_offset_mod
theta += th_offset

x1 = r*torch.cos(theta)
x2 = r*torch.sin(theta)

data = torch.stack([x1, x2], dim=1)
labels = torch.stack([A, B], dim=1).type(torch.long)
```

Listing 2: Code for toy DFR dataset

```python
import torch

def observational_points(num_obs_points, num_classes, support):
    Y1_obs = torch.randint(num_classes, size=(num_obs_points,))
    if support == "diff":
        Y2_obs = Y1_obs % (num_classes // 2)
    else:
        Y2_obs = Y1_obs.clone()
    Y_obs = torch.stack([Y1_obs, Y2_obs], dim=1)
    return Y_obs

def intervention_partial_support(num_classes, num_int_points):
    num_groups = num_classes // 2 # The classes are grouped into 4 groups
    Y1_int = torch.randint(num_classes, size=(num_int_points,))
    Y2_int = torch.empty_like(Y1_int)
    cl_per_gp = 2
    for _ in range(num_groups):
        mask = (cl_per_gp*_ <= Y1_int) & (Y1_int < cl_per_gp*(_+1))
        np = mask.sum().item()
        Y2_int[mask] = torch.randint(cl_per_gp*_, cl_per_gp*(_+1), size=(np,))
    Y_int = torch.stack([Y1_int, Y2_int], dim=1)
    return Y_int

def intervention_diff_support(num_classes, num_int_points):
    Y1_int = torch.randint(num_classes, size=(num_int_points,))
    Y2_int = torch.randint(num_classes // 2, size=(num_int_points,))
    Y_int = torch.stack([Y1_int, Y2_int], dim=1)
    return Y_int

def intervention_full_support(num_classes, num_int_points):
    Y1 = torch.randint(num_classes, size=(num_int_points,))
    Y2 = torch.randint(num_classes, size=(num_int_points,))
    Y_int = torch.stack([Y1, Y2], dim=1)
    return Y_int

def get_X_from_Y(Y, num_classes):
    mu_x1 = (1.1*Y[:, 0] - (num_classes - 1)/2.)
    mu_x2 = (1.1*Y[:, 1] - (num_classes - 1)/2.)

    X1 = mu_x1 - 0.5 + torch.rand_like(mu_x1)
    X2 = mu_x2 - 0.5 + torch.rand_like(mu_x2)
    X = torch.stack([X1, X2], dim=1)
    return X

beta = 0.5
num_points = 20000
inp_dim = 2
num_classes = 8
num_obs_points = int(beta * num_points)
num_int_points = (num_points - num_obs_points)

# Scenario 1: trn_support = "full"
# Scenario 2: trn_support = "partial"
# Scenario 3: trn_support = "diff"
trn_support = "full"

if trn_support == "full":
    int_fn = intervention_full_support
elif trn_support == "partial":
    int_fn = intervention_partial_support
elif trn_support == "diff":
    int_fn = intervention_diff_support
else:
    raise ValueError("Invalid trn_support")

Y_obs = observational_points(num_obs_points, num_classes, trn_support) # Create observational
    points
I_obs = torch.zeros(num_obs_points, dtype=torch.int)
Y_int = int_fn(num_classes, num_int_points) # Create interventional points
I_int = torch.ones(num_int_points, dtype=torch.int)
Y = torch.cat([Y_obs, Y_int], dim=0)
I = torch.cat([I_obs, I_int], dim=0)

# Create the observed data signal from the labels.
X = get_X_from_Y(Y, num_classes)
```

