# OpenReview forum: "Representation Learning from Interventional Data"
_ICLR.cc/2024/Conference — Submitted to ICLR 2024_

### Official Review · Reviewer_WWQo · 2023-10-30

**Soundness:** 3 good
**Presentation:** 3 good
**Contribution:** 2 fair
**Rating:** 5
**Confidence:** 4

**Summary:**

This paper first introduces a problem where the data is a mixture of observational and interventional data. The authors proposed a method that can limit the dependency between variables for interventional data based on NHSIC measure. Experiments demonstrate superior performance against current methods on synthetic datasets and other tasks.

**Strengths:**

Many case studies and examples are given to motivate the problem, and the results are clearly presented. The RepLin method seems to be a novel use of the NHSIC dependence measure. There are a lot of experiments spanning a variety of tasks including robustness to image corruption, which is very interesting. All experimental results demonstrated superior performance against benchmarks.

**Weaknesses:**

It seems that RepLin algorithm assumes knowledge of which datapoint is interventional, whereas the ERM and other methods do not use this assumption. ERM-Resampled uses that knowledge, but not as directly as RepLin. However, experimental comparisons with ERM-Resampled is absent. In Section 3, it is unclear what the author(s) are trying to discuss, especially the phrase “it is advisable to remove spurious information from the representations entirely”. The title of the paper is about representation learning, but the paper evaluates the quality of the representation only via a single downstream task, making the problem more like supervised learning.

**Questions:**

- What is the test dataset for the problem in Figure 3? If in Figure 3(b) the test is on the full support, then it is a significantly harder problem then Figure 3(a), making the motivation unclear.
- Why is there no comparison between RepLin and ERM-Resampled?
- Does the method work when there are more than two features that directly affect X?  What about when the intervention is on two variables?

---

> ### Author Response · Authors · 2023-11-16
> **Clarity on the method, results and terminology + additional results**
>
> **Comparison to ERM-Resampled:** Please refer to Table T1 for the comparison between RepLIn and ERM-Resampled.
>
> **Use of resampling:** We observed that resampling improves robustness in general. A comparison between our method without resampling and other methods can be found in Table T1. Additionally, resampling will help our method obtain better gradients from dependence-related losses as the estimate of the dependence improves with the number of samples.
>
> **Clarity for Section 3:** The phrase from Section 3 that the reviewer found unclear was intended to argue about the danger of leaving spurious information in the representation. In a recent line of work, it was proposed to fine-tune the classifier to achieve robustness after observing that the learned representation contained both invariant and spurious features [R12-15]. However, in the absence of a fine-tuning dataset that exactly matches the test distribution, some spurious correlations may still slip through. We demonstrate this in a case where the support of the interventional distribution does not match that of the test distribution (Sections 3 and 4.3 in the main paper).
>
> **Terminology in the title:** The larger goal of our work is to learn useful representations from interventional data. The term ``representation learning" is not limited to unsupervised/self-supervised learning. For example, representations learned from ImageNet classification have proved useful in several downstream applications.
>
> **Test dataset in Fig. 3:** In Fig. 3, $A$ and $B$ are binary variables. During observations, $B\coloneqq A$ and during interventions, $B$ is set to values independent of $A$. The training data consists of both observational and interventional data, so that the model may learn the true invariant features. Since we are interested in evaluating the robustness of our models to interventional distributional shifts, our test set is built entirely from interventional data.
>
> **Multi-variate case:** To demonstrate the advantage of our method in the multi-variate case, we construct a causal graph with five binary variables. The latents are generated according to the following SCM.
>
> $$
> A = \text{Bern}(0.4) \\
> $$
> $$
> B = \text{Bern}(0.6) \\
> $$
> $$
> C = A \vee B \\
> $$
> $$
> D = A \wedge C \\
> $$
> $$
> E = \neg B \wedge C
> $$
>
> where $\text{Bern}(p)$ indicates a Bernoulli distribution parameterized by $p$. The observed data $X$ is obtained as,
>
> $$
> X\_A = \text{NN}\_{4,3}(A)
> $$
> $$
> X\_B = \text{NN}\_{4,3}(B)
> $$
> $$
> X\_C = \text{NN}\_{2,0.1}(C)
> $$
> $$
> X\_D = \text{NN}\_{2,0.1}(D)
> $$
> $$
> X\_E = \text{NN}\_{2,0.1}(E)
> $$
> $$
> X = \text{concat}\left(X\_A, X\_B, X\_C, X\_D, X\_E\right)
> $$
>
> *where* $\text{NN}_{l,\sigma}$ indicates a randomly initialized neural network with $l$ hidden layers, each with 100 units, and an added noise layer which adds a Gaussian noise with zero mean and $\sigma$ standard deviation. Each latent variable adds information about itself to the observed signal. Our intuition here is that, due to more hidden layers and larger added noise, the model may struggle to use the invariant information from $X$ and instead learn spurious features.
>
> To collect interventional data, we intervene on $C$, $D$, and $E$ separately. Our evaluation metrics, therefore, will be the accuracy of the model in predicting $A$ and $B$ during these interventions. We compare ERM-Resampled and RepLIn-Resampled on their predictive accuracy during intervention on $C$, $D$, and $E$ in Tables T2, T3, and T4.

---

> > ### Author Response · Authors · 2023-11-16
> > **Results from multivariate experiments**
> >
> > Table T2-A: Predictive accuracy for $A$ during intervention on $C$.
> >
> > | Method | $\beta = 0.5$ | $\beta = 0.7$ | $\beta = 0.9$ | $\beta = 0.95$ |
> > | --- | --- | --- | --- | --- |
> > | ERM-Resampled | 91.94+\-0.19 | 92.17+\-0.14 | 91.99+\-0.32 | 91.91+\-0.53 |
> > | RepLIn-Resampled | 100.00+\-0.00 | 100.00+\-0.00 | 100.00+\-0.00 | 100.00+\-0.01 |
> >
> > Table T2-B: Predictive accuracy for $B$ during intervention on $C$.
> >
> > | Method | $\beta = 0.5$ | $\beta = 0.7$ | $\beta = 0.9$ | $\beta = 0.95$ |
> > | --- | --- | --- | --- | --- |
> > | ERM-Resampled | 93.64+\-1.00 | 96.31+\-1.50 | 96.46+\-1.80 | 95.50+\-1.07 |
> > | RepLIn-Resampled | 100.00+\-0.00 | 100.00+\-0.00 | 100.00+\-0.00 | 100.00+\-0.00 |
> >
> > Table T3-A: Predictive accuracy for $A$ during intervention on $D$.
> >
> > | Method | $\beta = 0.5$ | $\beta = 0.7$ | $\beta = 0.9$ | $\beta = 0.95$ |
> > | --- | --- | --- | --- | --- |
> > | ERM-Resampled | 74.18+\-2.36 | 73.96+\-1.35 | 71.81+\-1.19 | 73.34+\-2.68 |
> > | RepLIn-Resampled | 100.00+\-0.00 | 100.00+\-0.00 | 100.00+\-0.00 | 99.99+\-0.02 |
> >
> > Table T3-B: Predictive accuracy for $B$ during intervention on $D$.
> >
> > | Method | $\beta = 0.5$ | $\beta = 0.7$ | $\beta = 0.9$ | $\beta = 0.95$ |
> > | --- | --- | --- | --- | --- |
> > | ERM-Resampled | 98.57+\-0.17 | 99.45+\-0.38 | 98.50+\-1.18 | 98.14+\-0.90 |
> > | RepLIn-Resampled | 100.00+\-0.00 | 100.00+\-0.00 | 100.00+\-0.00 | 100.00+\-0.00 |
> >
> > Table T4-A: Predictive accuracy for $A$ during intervention on $E$.
> >
> > | Method | $\beta = 0.5$ | $\beta = 0.7$ | $\beta = 0.9$ | $\beta = 0.95$ |
> > | --- | --- | --- | --- | --- |
> > | ERM-Resampled | 100.00+\-0.01 | 100.00+\-0.00 | 100.00+\-0.00 | 99.97+\-0.06 |
> > | RepLIn-Resampled | 100.00+\-0.00 | 100.00+\-0.00 | 100.00+\-0.00 | 100.00+\-0.00 |
> >
> > Table T4-B: Predictive accuracy for $B$ during intervention on $E$.
> >
> > | Method | $\beta = 0.5$ | $\beta = 0.7$ | $\beta = 0.9$ | $\beta = 0.95$ |
> > | --- | --- | --- | --- | --- |
> > | ERM-Resampled | 92.07+\-0.13 | 91.76+\-0.15 | 92.10+\-0.26 | 91.90+\-0.34 |
> > | RepLIn-Resampled | 100.00+\-0.00 | 100.00+\-0.00 | 100.00+\-0.00 | 100.00+\-0.00 |

---

### Official Review · Reviewer_XFMF · 2023-10-30

**Soundness:** 2 fair
**Presentation:** 2 fair
**Contribution:** 2 fair
**Rating:** 3
**Confidence:** 3

**Summary:**

This paper considers the problem of representation learning under distribution shift. Specifically, the paper considers the setting where the data generating process is described by a causal DAG, and the learner is able to observe both observational data and data generated by a specific, known intervention. The paper shows that on a toy example, both ERM and a resampled-version of ERM fail to generalize when given a large amount of observational data but only a small amount of interventional data. The paper uses this example to motivate their representation learning approach which, in addition to minimizing predictive error, also seeks to minimize a statistical dependency measure between features that are known to be independent on the observational data. The paper then goes on to compare their method against ERM and the resampled version of ERM on a synthetic toy dataset and an image classification dataset where smiling and gender have been subsampled to induce dependence. The paper also compares against ERM and fine-tuning on an ImageNet subset where noise is added to correlate with certain labels. On all experiments, the paper shows that their method leads to improvement over the baselines.

**Strengths:**

The paper considers an important area of study: learning under distribution shift. The paper distills the particular type of distribution shift that they interested in into an easily understood model, and illustrates the issue with common heuristics in addressing this problem on a toy example. Finally, the paper considers both toy and realistic data in their experiments.

**Weaknesses:**

The paper has a few weaknesses that should be addressed.

1. It is unclear exactly where the dependency and self-dependency losses come from. In particular, they seem to be only heuristically motivated, and it's unclear what minimizers of this loss should look like. Why should the "self-dependence loss" prevent the model from learning only irrelevant features, and how do we know that this is the right loss to do so? A rigorous mathematical treatment of the proposed loss would be very helpful here.

2. The paper only considers ERM-based baselines. The field of domain generalization is currently rich with approaches to problems that subsume the current work, but this paper does not compare against any of those. E.g., invariant risk minimization is one approach to domain generalization that would seem to easily apply to the present problem.

3. The paper doesn't really motivate the problem setting. It's not clear to me when we would be in a setting where (a) we know the true causal DAG, (b) we get to observe interventional data, and (c) we want to learn representations of our data for some downstream purpose, e.g. prediction in this setting.

**Questions:**

Building on the weaknesses I pointed out above:

1. Can you show that minimizing the RepLin loss leads to provably optimal properties for the minimizing function?

2. Can you justify why you did not compare against methods from domain generalization?

3. Can you present a concrete setting that motivates the paper setting?

---

> ### Author Response · Authors · 2023-11-16
> **About loss functions**
>
> **Common responses:** The relevance and applications of our problem setting can be found in common response A. We discuss the difference between our goal and that in domain generalization in common response B.
>
> **Origin of dependence loss function:** Our dependence loss is motivated by the observed correlation between the drop in accuracy and feature dependence during interventions. Since dependence between learned features is commonly measured and adjusted using kernel-based measures of dependence such as HSIC and KCC, we use the same to design our loss function.
>
> **Origin of self-dependence loss function:** A caveat in learning independent features is that the presence of irrelevant/degenerate information in the representation can minimize the dependence loss while maintaining strong performance. To prevent this, we use our self-dependency loss which penalizes the representations for not fully aligned with its corresponding attribute of interest.

---

> > ### Comment · Reviewer_XFMF · 2023-11-21
> >
> > Thank you to the authors for the detailed responses to my concerns. I still have a couple of reservations about the paper in its present form, however.
> >
> > First, a common concern among the reviewers, myself included, is that the assumptions of the paper seem overly restrictive. In particular, perfect knowledge of the causal graph and perfect interventions are often not possible in interesting settings. For example, in the mRNA expression example given in the author response, the full biological causal graph may not be known ahead of time and the interventions are often noisy. I think the paper could be improved by demonstrating some robustness of their method to causal graphical misspecification.
> >
> > Second, I'm still not sure that I totally believe that minimizing the loss functions leads to the optimal causal representation. Is it possible for the authors to construct even a toy example where minimizing these loss functions in the high sample limit provably   leads to the optimal causal representation? Also, the appendix points out that there is a "warm-up" phase in the learning process where one of the regularization parameters is set to 0 and then gradually increased to its full value. Is this because the loss functions lead to unstable learning dynamics?

---

> > > ### Author Response · Authors · 2023-11-22
> > > **During imperfect interventions**
> > >
> > > **************************************************Imperfect interventions:************************************************** We do not require knowledge of the entire causal graph, but only the intervened targets and their parents.
> > >
> > > To evaluate our method on imperfect interventions, we modify the experiments on our Windmill data to model imperfect interventions. Specifically, we fix the proportion of observational data $\beta$ to 0.5 and we add a new parameter $\gamma$ that denotes the chance of an intervention being perfect. Our method does not assume the knowledge of $\gamma$ and enforces its loss on this imperfect interventional data.
> > >
> > > The results comparing our method against ERM-Resampled are shown below in Table T5. The performances of both Replin-Resampled and ERM-Resampled drop, but our method still outperforms the baseline. We believe that the performance drop in our method is due to wrong independence relations being enforced, while that in the baseline is due to a general decrease in the diversity of the data (higher $\gamma$ is equivalent to lower $\beta$).
> > >
> > > Table T5: Predictive accuracy for $A$ on Windmill dataset under varying proportions of imperfect interventional data.
> > >
> > > | Method | $\gamma = 0.1$ | $\gamma = 0.2$ | $\gamma = 0.4$ | $\gamma = 0.6$ | $\gamma = 0.8$ | $\gamma = 0.9$ |
> > > | --- | --- | --- | --- | --- | --- | --- |
> > > | ERM-Resampled | 63.69 +\- 0.74 | 63.41 +\- 1.27 | 60.32 +\- 0.37 | 57.07 +\- 0.81 | 50.22 +\- 0.57 | 49.89 +\- 0.50 |
> > > | RepLIn-Resampled | 89.71 +\- 4.6 | 88.18 +\- 1.8 | 80.54 +\- 4.67 | 68.34 +\- 5.41 | 63.30 +\- 5.69 | 60.31 +\- 0.25 |
> > >
> > > ********************************************Optimal causal representations:******************************************** As mentioned the earlier responses, our objective is not to learn identifiable causal representations. Indeed, there is no guarantee that our representations will be identifiable. Our objective is to provide robustness against interventional distribution shifts. As interventional distribution shifts cannot be forecast without experimental data collection, we assume the knowledge of the interventional target and the parents of the intervened variable during training.
> > >
> > > **************************Warm-up phase:************************** Our ultimate goal is to achieve good predictive performance that is robust to interventional distribution shifts. We observed that strongly enforcing our loss functions, particularly the dependence loss function, in the early stages of training hindered the model from learning useful representations. Warming up ensures that the representations remain useful while the independence loss functions are minimized.

---

### Official Review · Reviewer_Uyzd · 2023-10-31

**Soundness:** 3 good
**Presentation:** 4 excellent
**Contribution:** 1 poor
**Rating:** 3
**Confidence:** 4

**Summary:**

The paper considers the problem of prediction under distribution shifts as a result of interventions on one of the label and proposes an approach inspired by the conditional independence in the intervened graph. The authors consider a setup with multiple labels with causal relationships between the labels, and known hard-interventions that act on one of the labels to remove its dependence from other relevant labels. The proposed approach enforces statistical independence between intervened labels and their parent labels in the original graph; and highlights its benefits over several baselines like reweighting, classifier fine-tuning, etc. on multiple benchmarks.

**Strengths:**

* The claims made in the paper are supported by rigorous experimentation over multiple synthetic, semi-synthetic, and realistic benchmarks. Further, the authors also compare with important baselines of reweighting, classifier fine-tuning, pre-training, etc.

* The paper is well-written with good motivation for the proposed approach using synthetic benchmarks, and the presentation of the results is clear and well-organized.

* The proposed approach uses loss objectives from prior work to enforce independence between features with minor modifications, though the application for the task of multi-label prediction with interventions is novel.

**Weaknesses:**

* My major concern with the work is that the technical contribution of the work is weak since the problem setup is very simple. The authors assume they know exactly the label that has been intervened and its non-descendants, which makes the proposed approach a simple application of the d-separation criteria in causal graphs. In contrast, the common setups in OOD generalization involving distribution shifts [1, 2] do not assume that we observe the intervened variables (and their causes) explicitly.

* I appreciate the rigorous experimentation in the work, however, the significance of the results is highly limited as the application of d-separation criteria in the intervened graph with known variables and causal relationships is supposed to work. Further, the experiments on the corrupted CIFAR and ImageNet require the knowledge of corruption labels, which would simplify the problem.

References:

[1] Martin Arjovsky, Léon Bottou, Ishaan Gulrajani, and David Lopez-Paz. Invariant risk minimization.
arXiv preprint arXiv:1907.02893, 2019

[2] Polina Kirichenko, Pavel Izmailov, and Andrew Gordon Wilson. Last layer re-training is sufficient
for robustness to spurious correlations. arXiv preprint arXiv:2204.02937, 2022

**Questions:**

My suggestion to the authors is to consider the case with unknown causal relationships and intervened nodes, as that is a more practical and challenging setup. I understand that would require substantial efforts with a completely different approach in principle to solve the problem, but I am interested in hearing the thoughts of the authors regarding this point.

Minor points:

* The description of the prior work Ahuja et al. (2022) in the learning using interventional data section is incorrect. Their work is not limited to independent latent factors, rather they allowed for general causal relationships between the latents.

---

> ### Author Response · Authors · 2023-11-16
> **Approach and Related Works**
>
> **Common responses:** The existing works on OOD generalization are designed for general distribution shifts. However, we are interested in a specific form of distribution shift induced by causal interventions. Please refer to common response B, where we describe how our goals and approach differ from standard invariant learning.
>
> **Our approach and d-separation:** d-separation only tells us what the true dependence relations in the ideal causal model are. It does not translate to learning systems where no checks are employed during training to enforce true causal relations. One of our contributions is the observation that d-separation is not inherently followed by these models. Our method was driven by the observed correlation between the drop in accuracy and dependence between the features during interventions, as discussed in Section 2.1 of our paper. This observation is non-trivial and has not been previously discussed in the literature on learning from interventional data, although the question of whether learned models respect causal relationships has been asked [R21-23].
>
> **Knowledge of corruption labels:** Robustness to label-dependent corruption is one of the several applications of our work. As we mentioned in common response A, we may often have this information. In practice, it is possible to infer the type of image corruption [R11].
>
> **Correction in related works:** We apologize for the error in the description of Ahuja et al. (2022). Their paper indeed says it “does not require any structural assumptions about the dependency between the latents”. We have made this correction in the newly added section that contrasts our setting with that of identifiable causal representation learning.

---

> ### Comment · Reviewer_Uyzd · 2023-11-21
>
> **Common Response B**
>
> Regarding the common response B, I'm afraid I have to disagree that datasets like PACS are not a result of intervention. Changes in the style of the image (photo, sketch, etc.) can be attributed to intervening on the latent variables that control to the style variations in the image. The goal of domain generalization methods is more difficult since they do not observe these latent variables and the causal graph for the data generation process, but the proposed approach in this work relies heavily on them. I do understand the difference between the proposed setup and the domain generalization setup, but the former does not seem challenging due to the reasons I mentioned before.
>
> **Comparison with related works from Causal Representation Learning**
>
> After reading the common response A, perhaps a case could be made that assuming the knowledge of intervened variables can be practical in some cases. But then a proper comparison needs to be made with other approaches from causal representation learning that can do so. For example, some works enforce support independence via Hausdorff distance [1, 2] between latent representations, which is a richer penalty than enforcing statistical independence as it can handle the case of correlation due to unobserved confounders as well. The proposed setup of authors can be been similar to the work of Ahuja et al. [2], the only difference being that the authors consider a supervised setup with a classification loss; while the setup of Ahuja et al. is unsupervised and they have a reconstruction objective. However, the access to interventional data is similar and the penalty of enforcing support distance in Ahuja et al. can be used by the authors in the proposed setup as well.
>
> I do agree with the authors when they say: "The general objective in identifiable causal representation learning is to “learn both the true joint distribution over both observed and latent variables. The objective of this work is to provide a method to learn representations that are robust to interventional distribution shifts under the assumption of known interventional targets and their parents."
>
> However, there are rich ideas in the causal representation learning about enforcing (statistical/support) independence, etc, which can serve as competitive baselines in the proposed setup.
>
> I do appreciate the effort made by the authors but at this time the submission indeed requires more work and unfortunately I would not be able to increase my score. In the current form, the authors have technically sound results but I think the overall contribution is limited. I encourage authors to incorporate my feedback about causal rep learning baselines since I think it can make a strong case for downstream applications of the various causal rep learning works that have mostly dealt with identification and small-scale experiments for now.
>
> References
>
> [1] Roth, Karsten, Mark Ibrahim, Zeynep Akata, Pascal Vincent, and Diane Bouchacourt. "Disentanglement of correlated factors via hausdorff factorized support." arXiv preprint arXiv:2210.07347 (2022).
>
> [2] Ahuja, Kartik, Divyat Mahajan, Yixin Wang, and Yoshua Bengio. "Interventional causal representation learning." In International conference on machine learning, pp. 372-407. PMLR, 2023.

---

### Official Review · Reviewer_Pdav · 2023-11-01

**Soundness:** 2 fair
**Presentation:** 3 good
**Contribution:** 1 poor
**Rating:** 3
**Confidence:** 3

**Summary:**

In this work the authors propose the RepLIn method for representation learning that aims to be more robust under distribution shift obtained via a perfect intervention on a single feature. For a model that has a causal data generating process, standard techniques like ERM propose to learn the generative process and the recent field of causal representation learning has brought along an array of sophisticated tools to handle this. In general, when the data generating process is modified by a hard intervention on a causal variable, this brings a distribution shift and the learned representations are not robust. The authors propose to modify the loss function of representation learning methods using their regularization term to handle such issues.

Initial experiments on the windmill datasets (with 2 causal variables) suggest that there is a correlation between accuracy drop on interventional data and independence of the features corresponding to the intervened variable. This situation is somewhat mitigated by using additional interventional data. However, the authors propose a different fix which is to define a modified regularization term that aims to minimize the dependence between features on interventional data. To avoid pathological feature learning, an additional regularization term is added. The authors call this the RepLIn method.

They validate their observation of correlation on the windmill dataset. Experiments on CelebA dataset (using the causal variables smile and gender) show that RepLIn is more robust than the baseline algorithms by 2%. Additional experiments on 3-variable causal data generating process is also shown on label prediction. These experiments weakly validate the method and suggest that spurious correlations are mitigated.

### References:

- [1] Interventional causal representation learning.

- [2] Learning Linear Causal Representations from Interventions under General Nonlinear Mixing

**Strengths:**

- The statistical correlation between drop in representation learning accuracy and independence of the causal covariates is a nice observation. Retrospectively, it's not surprising considering the data generating process but the authors suggest to exploit this via explicit regularization.

- Experiments show that the proposed method RepLIn can beat methods such as classifier fine-tuning to handle distribution shift on various image datasets.

**Weaknesses:**

- The kind of distribution shift considered, i.e. hard perfect interventions on causal features, maybe too restrictive for real-life applications. In fact, IRM aims to be as general as possible, unfortunately at the overhead of significant additional complexity. Therefore, this model loses this generality.

- Related to the above, even if we have weak dependence among the causal variables after intervention, the proposed regularization method will still be feasilbe to apply, but it's not clear how much the hyperparameter tuning will be needed in the experiments. Could the authors comment on this?

- The authors should have a detailed discussion on the array of 10+ works on causal representation learning from interventional data, e.g. [1], [2] (see also related works section in [2] which contains a wide list of such works). I'm surprised the authors seem unaware of this very closely related line of work.

- Related to the above point, while the authors compare to IRM in app. F and highlight the differences, I feel like the works on causal representation learning are more apt to contrast with, due to various modeling assumptions.

**Questions:**

Some questions were raised above.

- The authors consider perfect interventions. How robust are their methods to imperfect or soft interventions (with the usual meanings from the causal literature)?

---

> ### Author Response · Authors · 2023-11-16
> **Assumptions and Relevance of our Problem Setting**
>
> **Common responses:** We justify the relevance and practicality of our setting in common response A. We apologize for the confusion due to the term "causal representation learning,” which is more commonly used in the identifiable learning community. Our response to your questions about the missing related works from identifiable causal representation learning can be found in common response C.
>
> **About assumptions:** The primary objective of our work is to develop a method that can efficiently exploit the limited amounts of interventional data that may be available for training. To this end, we make a few assumptions. However, our assumptions are justified given that our work is the first to explicitly exploit interventions to achieve robustness to interventional distribution shift.
>
> **Imperfect/soft interventions:** We developed our method under the assumption of perfect interventions. Constraining the dependence between intervened variables can be posed as a constrained optimization problem for which closed-form solutions may be obtained. For example, [R10] models the allowed sensitive information leakage as a constraint and obtains a closed-form solution. Nonetheless, we believe minimizing dependence between the interventional features can improve robustness against such distribution shifts.

---

> > ### Comment · Reviewer_Pdav · 2023-11-21
> >
> > I thank the authors for their response. Rereading the work in light of the other reviews, I also agree with the other reviewers that the contributions are very limited (given the strong assumptions made) and the writing needs to be significantly improved.

---

### Author Response · Authors · 2023-11-16
**Common Response**

We thank all the reviewers for their kind and valuable comments. We greatly appreciate the insights they have shared and are sure it improves the quality of our work. We have modified the original PDF in response to the comments and highlighted the changes in violet. Here, we provide our response to the questions that multiple reviewers raised. Please consider increasing your scores if we have answered your concerns. If you have further questions or concerns, we would be very happy to answer them.

## A. Relevance of the problem setting

A common concern shared by all reviewers was the relevance of the problem setting in our submission. As **XFMF** summarized, we consider the setting where (a) we know the true underlying causal DAG, (b) we get to observe interventional data, and (c) we want to learn representations for some downstream purpose. Below, we detail a few possible applications:

1. **************************************Object Recognition:************************************** Learning object recognition models invariant to spurious factors such as background, domain, and other objects in the scene [R1-3]. Matching this application to our setting: (a) the spurious factor is treated as a variable that shares a common cause with the object label or is caused by the label [R17-18], (b) interventional data is obtained manually or through generative models [R4], and (c) the objective is to learn models that are robust to this interventional distribution shift.
2. ************************************Gene Expressions Measurement [R19]:************************************ Data was collected from interventions through gene deletions [R20] and was used to train models (GNNs and CNNs) for gene expression measurement.
3. ****************************Large Models:**************************** As large models such as CLIP, GPT-3, and ViT-22B become ubiquitous in industry and academia, we envision scenarios where it would become necessary to improve or correct them using minimal amounts of experimentally collected interventional data. We believe that the fundamental idea in our work, namely, how to exploit your interventional data, will be relevant in such scenarios.

## B. Relation from Standard Invariant Learning

Another common question raised by the reviewers was the relation between our motivation and standard invariant learning, which includes domain generalization, mitigating spurious correlations, and algorithmic fairness to minority groups. Reviewers also pointed out relevant approaches, such as IRM [R6]. In invariant learning, the objective is to learn an invariant predictor from a training set that comprises data passively collected from different groups/environments. The underlying assumption is that a truly invariant predictor can only use features that do not change with the environment. Section 4 of [R6] mentions that each intervention can generate a new environment under some assumptions. However, the corollary is not true: interventions are not required to generate a new environment. For example, consider the images in the PACS dataset for domain adaptation [R16], which were collected from domains such as sketches and paintings without any intervention. Therefore, we argue and demonstrate below that methods that leverage the diversity of data from multiple environments may not be the optimal way to learn from interventional data.

In contrast to methods like IRM and DRO, which were designed to leverage the general diversity in the data, our method exploits the available interventional data using our prior knowledge of the causal graph. The resulting advantage of our method can be seen in Table T1, where we compare our method against common baselines like IRM [R6], DRO [R5], and Fish [R8].

## C. Comparison to Identifiable Causal Representation Learning

We thank the reviewers for mentioning the works on identifiable learning. We wish to correct our use of the term “causal representation learning” as it is more commonly used in the identifiable causal representation learning (ICRL) community, whose goal is to learn a representation, typically using an autoencoder, such that the decoder for this representation implicitly models the true underlying SCM. The primary objective of ICRL is to learn the joint distribution of the observed and the latent variables. However, our objective is not to learn this joint distribution but instead propose a method that can explicitly account for the interventional data available during training. We have explained the relevance of our objective along with some applications in common response A. We have added a section in the appendix of our paper (App G, highlighted in violet) where we categorize the works on ICRL, specifically discuss those that use interventional data, and then describe the fundamental difference between our objective and ICRL.

---

> ### Author Response · Authors · 2023-11-16
> **Additional Comments and References**
>
> ## Additional Results
> Table T1: Comparison of our method against different baselines on the proposed Windmill dataset (Section 2.1 in the main paper)
>
> | Method | $\beta=0.5$ | $\beta =0.7$ | $\beta = 0.9$ | $\beta=0.95$ |
> | --- | --- | --- | --- | --- |
> | ERM | 76.42 +/- 3.74 | 61.49 +/- 2.54 | 50.09 +/- 0.38 | 50.03 +/- 0.29 |
> | ERM-Resampled | 65.40 +/- 1.81 | 65.71 +/- 2.02 | 69.39 +/- 3.57 | 69.37 +/- 1.65 |
> | IRM [R6] | 75.23 +/- 1.63 | 78.21 +/- 1.37 | 78.28 +/- 1.01 | 78.31 +/- 1.65 |
> | DRO [R7] | 82.21 +/- 1.57 | 84.97 +/- 1.05 | 85.76 +/- 0.51 | 84.33 +/- 1.32 |
> | Fish [R8] | 63.93 +/- 0.96 | 64.76 +/- 0.98 | 64.32 +/- 1.25 | 67.35 +/- 2.56 |
> | RepLIn | 91.72 +/- 2.72 | 77.22 +/- 2.03 | 62.55 +/- 3.41 | 55.68 +/- 4.88 |
> | RepLIn-Resampled | 90.04 +/- 2.85 | 91.85 +/- 2.61 | 93.79 +/- 1.69 | 92.20 +/- 0.72 |
>
> ## References
>
> [R1] Dunlap et al., “Diversify Your Vision Datasets with Automatic Diffusion-Based Augmentation”, NeurIPS 2023
>
> [R2] Azizi et al., "Synthetic Data from Diffusion Models Improves ImageNet Classification,” TMLR 2023
>
> [R3] Gao et al., "Out-of-Domain Robustness via Targeted Augmentations,” ICML 2023
>
> [R4] Sauer and Geiger, "Counterfactual Generative Networks,” ICLR 2021
>
> [R5] Idrissi et al., "Simple data balancing achieves competitive worst-group-accuracy,” CLeaR 2022
>
> [R6] Arjovsky et al., "Invariant Risk Minimization,” ArXiv 2019
>
> [R7] Sagawa et al., "Distributionally Robust Neural Networks for Group Shifts,” ICLR 2020
>
> [R8] Shi et al., "Gradient Matching for Domain Generalization,” ICLR 2022
>
> [R9] Nguyen et al., "Quality Not Quantity: On the Interaction between Dataset Design and Robustness of CLIP,” NeurIPS 2022
>
> [R10] Sadeghi et al., "On the Global Optima of Kernelized Adversarial Representation Learning,” ICCV 2019
>
> [R11] Li et al., "All-In-One Image Restoration for Unknown Corruption,” CVPR 2022
>
> [R12] Menon et al., "Overparameterisation and Worst-Case Generalisation,” ICLR 2021
>
> [R13] Kirichenko et al., "Last Layer Re-Training is Sufficient for Robustness to Spurious Correlations,” ICLR 2023
>
> [R14] Rosenfeld et al., "Domain-Adjusted Regression or: ERM May Already Learn Features Sufficient for Out-of-Distribution Generalization,” NeurIPS DistShift 2022
>
> [R15] Qiu et al., "Simple and Fast Group Robustness by Automatic Feature Reweighting,” ICML 2023
>
> [R16] Li et al., "Deeper, Broader and Artier Domain Generalization,” ICCV 2017
>
> [R17] Ilse et al., “Selecting Data Augmentation for Simulating Interventions”, ICML 2021
>
> [R18] Mitrovic et al., “Representation Learning via Invariant Causal Mechanisms”, ICLR 2021
>
> [R19] Lagemann et al., “Deep learning of causal structures in high dimensions under data limitations”, Nature Machine Intelligence 2023
>
> [R20] Kemmern et al., “Large-Scale Genetic Perturbations Reveal Regulatory Networks and an Abundance of Gene-Specific Repressors”, Cell 2014
>
> [R21] Jin et al., “Can Large Language Models Infer Causation from Correlation?”, ArXiv 2023
>
> [R22] Wang and Boddeti, “Do Learned Representations Respect Causal Relationships?”, CVPR 2022
>
> [R23] Kıcıman et al., “Causal Reasoning and Large Language Models: Opening a New Frontier for Causality”, ArXiv 2023

---

### Author Response · Authors · 2023-11-20
**A Gentle Reminder to All the Reviewers**

We appreciate the effort that the reviewers have put into providing valuable feedback on our work. We wish to gently remind all the reviewers that the rebuttal session will close in less than three days. We would be happy to answer any further questions that any reviewer has after reading our rebuttal.

---

### Author Response · Authors · 2023-11-23
**Thanks to all the reviewers**

Before the rebuttal session ends, we would like to thank all the reviewers for their thorough and sincere comments. We deeply appreciate their time and commitment to the research community. We promise to revise our work and incorporate the key points raised by the reviewers.

---

### Meta-Review · Area_Chair_9DjA · 2023-12-08

**Metareview:**

The authors propose a method for learning representations via hard perfect interventions. Compared to existing results in the literature, the results in this paper make stronger assumptions and thus are somewhat straightforward to derive. Moreover, all reviewers felt the problem setting was difficult to imagine having practical applications. While this isn't itself an issue, the lack of a clear theoretical contribution compared to known results is. So it is not clear what new insights have been provided.

**Justification For Why Not Higher Score:**

See meta-review

**Justification For Why Not Lower Score:**

N/A

---

### Decision · Program_Chairs · 2024-01-16

Reject